## Registered report

ecology/environmental science

endangered species, poaching, policy signal, survival analysis, large carnivore, *Canis rufus*

**Author for correspondence:**
Francisco J. Santiago-Ávila
e-mail: fj.santiagoavila@gmail.com

# Evaluating how management policies affect red wolf mortality and disappearance

Francisco J. Santiago-Ávila[1,2,3], Suzanne Agan[4], Joseph W. Hinton[5] and Adrian Treves[1]

[1]Nelson Institute for Environmental Studies, University of Wisconsin, Madison, WI, USA
[2]Project Coyote, Larkspur, CA, USA
[3]The Rewilding Institute, Albuquerque, NM, USA
[4]Department of Ecology, Evolution, and Organismal Biology, Kennesaw State University, Kennesaw, GA, USA
[5]Wolf Conservation Center, South Salem, NY, USA

FJS-Á, 0000-0003-4233-9128; AT, 0000-0002-3052-4708

Poaching is the major cause of death for large carnivores in several regions, contributing to their global endangerment. The traditional hypothesis used in wildlife management (killing for tolerance) suggests reducing protections for a species will decrease poaching. However, recent studies suggest reducing protections will instead increase poaching (facilitated illegal killing) and its concealment (facilitated cryptic poaching). Here, we build survival and competing risk models for mortality and disappearances of adult collared red wolves (*Canis rufus*) released in North Carolina, USA from 1987 to 2020 (*n* = 526). We evaluated how changes in federal and state policies protecting red wolves influenced the hazard and incidence of mortality and disappearance. We observed substantial increases in the hazard and incidence of red wolf reported poaching, and smaller increases in disappearances, during periods of reduced federal and state protections (including liberalizing hunting of coyotes, *C. latrans*); white-tailed deer (*Odocoileus virginianus*) and American black bear (*Ursus americanus*) hunting seasons; and management phases. Observed increases in hazard (85–256%) and incidence of reported poaching (56–243%) support the 'facilitated illegal killing' hypothesis. We suggest improving protective policies intended to conserve endangered species generally and large carnivores in particular, to mitigate environmental crimes and generally improve the protection of wild animals.

## 1. Background

Humans are responsible for most large carnivore deaths, worldwide [1] as in the USA [2–4]. Human practices have

triggered the decline and extirpation of carnivore populations through direct killing and the commonly associated threats of prey depletion, habitat loss and degradation [5]. In turn, the decimation of carnivore populations and loss of top-down regulation has contributed to biodiversity loss, simplification of trophic structures and the degradation of ecosystems and their functions [5–7]. Many management policies for large carnivores focus on regulating anthropogenic mortality, which hinges on robust scientific evaluations of the effects of said policies on direct carnivore killing. Given the scientific uncertainties with the sustainability of killing carnivores [4,5,8], important conservation outcomes such as population recovery, preservation of rare genotypes or behavioural traditions of individuals of endangered species, long-term viability of small remnant populations can be hindered by the effects of policies that reduce individual survival as well as by bias inherent in mortality analyses that commonly censor the full information gained from the disappearance of marked animals or dismiss the effects of policy believed to be helpful [4,8–11].

Illegal killing (poaching hereafter) is the major cause of anthropogenic mortality for many large carnivore populations [3,12–14]. Poaching can undermine conservation efforts by slowing population growth [15–17], removing monitoring data from marked animals whose remains and transmitters are destroyed [18–20], and hindering recolonization of the species' historic range [12,13]. Poaching's illicit nature makes it extremely difficult to identify, estimate and prevent. The concealment of activity and evidence means many monitored, poached animals are never detected, a phenomenon referred to in the scientific literature as 'cryptic poaching' [12]. Accurate estimates of policy and management effects on total poaching (the sum of reported and 'cryptic') are critical for improving the protection of any threatened or endangered species and are prominent in concerns over the conservation of large carnivores. However, the concealment of evidence that usually accompanies cryptic poaching complicates accurate estimation with systematic underestimates [3,12,21]; in places where poaching is often cryptic, decision-makers may misidentify the major threats and management interventions may be designed poorly to achieve conservation goals [3,12,21].

Scientific accounting for cryptic poaching has been facilitated by statistical advances that employ previously dismissed data to correct for its censoring and the resulting systematic underestimation of hazard and incidence of poaching. A common assumption in traditional analyses is that marked individuals that disappeared from monitoring suffer from the same fates (and at the same rates) as those individuals found dead. However, the occurrence of cryptic poaching means disappearances and other forms of uncertainty (such as detection probability) disproportionately affect poaching relative to other causes of death such as legal or regulated killing, which should be perfectly reported if it is indeed regulated or legal [4]. In the case of grey wolves (Canis lupus) and grey wolf populations, this erroneous assumption has led to systematic underestimates of poaching and overall mortality rates, and a failure to effectively intervene to mitigate anthropogenic killing [3,10,11,21]. For other wolf and large carnivore populations, questions remain over how management policies that increase or reduce protections for a species mediate poaching and overall mortality rates generally.

Few studies to date have considered if and how policy interventions affect the hazard and incidence of poaching while accounting for disappeared individuals and cryptic poaching (table 1, columns A and B). The traditional, and still pervasive, assumption in wildlife management is that some policy interventions liberalizing killing (through reduced protections, such as permits for killing) may increase human tolerance for a species or approval of management policies, leading to a reduction in overall poaching. Indeed, despite contrary evidence from multiple sources since 2013, the US Fish and Wildlife Service (USFWS) has argued in favour of this 'killing for tolerance' (table 1, column B, summarized in [22]) hypothesis in federal court and as recently as December 2020 in intergovernmental communication [23]. Early tests of this hypothesis from Wisconsin attributed poaching rates and population changes to intolerance motivated by management inconsistencies and controversial wolf policies (such as increased protections) [24,25]. On the other hand, Chapron & Treves [15] proposed the 'facilitated illegal killing' hypothesis (table 1, column B) to explain reported slow-downs in wolf population growth during alternating periods of policy interventions that reduced protections for grey wolves in Wisconsin and Michigan, USA. The latter authors inferred that cryptic poaching was responsible for the observed population slow-downs, along with a policy effect such that reducing protections for a species may either decrease the value of those individuals to potential poachers or decrease the risk of detection [15]. Social science studies looking at the same alternating wolf policy periods in Wisconsin found tolerance decreased and intentions to poach grey wolves increased after policy interventions that reduced protections for wolves [26–28]. In Scandinavia, Suutarinen & Kojola [29] found that increasing the number of grey wolves that can be killed legally was associated with an increase in poaching risk, while the actual number of legally hunted wolves was associated with a decrease in poaching risk. The most recent study on the

**Table 1.** Relationship between our hypotheses, proposed analyses and interpretation of outcomes (including contingent interpretation and synthesis of model results). Cox PH refers to Cox proportional hazards models, while FG refers to competing risk models. $HR_{poa}$ refers to the HR of the reported poaching endpoint, while $HR_{ltf}$ refers to the HR of the LTF endpoint (table modified from table 4 in [10]).

| A. question | B. hypotheses | C. analysis plan | D. interpretation given different outcomes[a] |
|---|---|---|---|
| does the hazard or incidence of death by reported poaching or disappearance of wild, collared adult red wolves change after policies change from more to less protection and back again? | 'killing for tolerance' predicts the hazard and incidence of reported poaching (POA) and disappearances (LTF) decline when reducing protections for the species. | Cox PH models (for each endpoint) on policy and individual covariates | ($HR_{poa}$ has to be less than 1 and greater in magnitude than any increase in $HR_{ltf}$ OR $HR_{ltf}$ has to be less than 1 and greater in magnitude than any increase in $HR_{poa}$) AND |
| | | competing risk FG models (for each endpoint) on policy and individual covariates | endpoint-specific CIFs estimate which endpoint has a greater effect on the population |
| | | CIFs allow for analysis of population effects (incidence) while considering the prevalence of each endpoint in the population | the criterion for determining if TOTAL poaching probability for wolves declined is a decline in the combined incidence of the LTF and POA endpoints |
| | 'facilitated illegal killing' predicts the hazard and incidence of reported poaching (POA) and disappearances (LTF) increase when reducing protections for the species. | Cox PH models (for each endpoint) on policy and individual covariates | ($HR_{poa}$ has to be greater than 1 and greater than any decrease in $HR_{ltf}$ OR $HR_{ltf}$ has to be greater than 1 and greater than any decrease in $HR_{poa}$) AND |
| | | competing risk FG models (for each endpoint) on policy and individual covariates | endpoint-specific CIFs estimate which endpoint has a greater effect on the population (from FG models of competing risks) |
| | | CIFs allow for analysis of population effects (incidence) while considering the prevalence of each endpoint in the population | the criterion for determining if TOTAL poaching probability for wolves increased is an increase in the combined incidence of the LTF and POA endpoints |
| | 'facilitated cryptic poaching' predicts the hazard and incidence increase for the endpoint LTF when reducing protections for the species. | Cox PH models (for LTF endpoint) on policy and individual covariates | $HR_{ltf} > 1$ AND |
| | | competing risk FG models (for LTF endpoint) on policy and individual covariates | endpoint-specific CIFs estimate the effect on the population (from FG models of competing risks) |
| | | LTF CIFs allow for analysis of population effects (incidence) while considering the prevalence of each endpoint in the population | |

[a]BF estimate the strength for our alternative and null hypotheses for each endpoint of interest, as well as assess inconclusiveness of the data.

Scandinavian wolf population found that higher legal removal of breeding pairs was associated with fewer such animals disappearing (potentially poached) [30], but concerns remain over their statistical analyses and handling of disappearances during years with legal wolf-killing [31].

Scientists can directly test the above hypotheses on the relationship between changing protections and poaching of wolves by analysing wolf mortality while accounting for (i) timing and duration of each wolf's exposure to periods with different levels of protection and (ii) using all available data by including wolf disappearances in time-to-event analyses. The latter step is essential to contending with cryptic poaching. Moreover, following [10], the 'killing for tolerance' hypothesis implies greater acceptance of wolves on the landscape, which may affect other sources of mortality besides poaching, such as legal killings or final agency removals prompted by private landowners. Therefore, fully addressing the opposed hypotheses above demands an examination of individual life histories and the interactions between different causes of death and disappearance.

Two recent studies have applied such methods to this question, for two US wolf populations. Santiago-Ávila *et al.* [11] reanalysed data on monitored Wisconsin, USA grey wolves (1979–2012) and found periods of reduced federal protections for wolves were associated with increases in both the hazard and incidence of wolf disappearances, which over-compensated for associated, minimal decreases in reported poaching. That study hypothesized for the first time that loosening endangered species protections drove poaching underground from reported to cryptic or altered reporting rates, a hypothesis we refer to as 'facilitated cryptic poaching' (table 1, column B). Additionally, legal killing of wolves by agency personnel in response to complaints increased with time during reduced protection periods, further undermining any claims to 'tolerance' among actors who could call on the state to kill wolves legally [11]. Next, Louchouarn *et al.* [10] analysed deaths and disappearances of endangered Mexican grey wolves (*C. l. baileyi*) in Arizona and New Mexico, USA (1998–2016) under even more controlled conditions: the hazard and incidence of government-implemented removals of Mexican wolves did not change during the same periods, eliminating any potentially confounding effects of super-additive mortality and undermining the notion that the government reduced protections so they could act (i.e. remove wolves) to raise tolerance [10]. They found periods of reduced federal protections for the subspecies were associated with increases of 121% in the hazard of disappearances. As in the Wisconsin study [11], the disappearances over-compensated for minimal declines in incidence of reported poaching. Therefore, Louchouarn *et al.* [10] dismissed the 'killing for tolerance' hypothesis and added support in favour of both the 'facilitated illegal killing' (increased poaching) and 'facilitated cryptic poaching' (increased *cryptic* poaching) hypotheses. Indeed, the latter study suggests that a government policy signal without attendant government action to remove or kill wolves sufficed to elevate poaching and shift it from reported to cryptic.

Here, we build on past research by testing all three hypotheses, 'killing for tolerance', 'facilitated illegal killing' and 'facilitated cryptic poaching', about the relationship between reduced protection periods with the mortality and disappearance of red wolves (*C. rufus*) in northeastern North Carolina, USA (table 1, columns A–D). We propose a new hypothesis that interventions in federal or state policies that exposed red wolves to more anthropogenic killing without explicitly loosening the endangered species protections would suffice to promote poaching. We test it against the alternative that hazard and incidence of poaching (reported or cryptic) did not change over time. Moreover, we explore if and how each component of total poaching, reported and cryptic, changes with policy interventions.

The history of reintroduced red wolves is essential to understanding our hypotheses. Red wolves are native to the temperate forest region of the eastern USA and were once the dominant carnivore in that region before they were largely exterminated by intensive predator control programmes and hunting [32]. The USFWS established the Red Wolf Recovery Program (recovery programme) following the species' listing under the Endangered Species Act (ESA) and its 1966 precursor [32,33]. Given the unfeasibility of *in situ* restoration, the recovery programme intentionally attempted to extirpate the population from the wild by capturing remaining individuals, instead focusing on capture for propagation in captivity [32,33]. The species was declared extinct in the wild by 1980 [33]. The development of the captive red wolf breeding programme allowed for the reintroduction of red wolves into Alligator River National Wildlife Refuge (ARNWR) in northeastern North Carolina, USA beginning in 1987 [32]. Since coyote expansion into the ARNWR in the early 1990s, the recovery programme has largely focused on (mitigating) hybridization with coyotes as the main threat to the species' recovery [34–37]. Once hybridization was successfully managed [38] and established as a consequence of anthropogenic mortality [39,40], concerns shifted to understanding red wolf mortality and specifically strategies to mitigate anthropogenic causes, given evidence of increased gunshot deaths since 1999 [35–37,40–45]. The reintroduced population initially grew steadily, peaking at

around 150 individuals in 2005 but had declined to 45–60 by 2016 [41] and to 7 wolves by 2020 (table 1 in [46]). The dire status of this critically endangered population underscores the importance of our test; especially considering the multiple, controversial policy interventions associated with reduced protections for red wolves.

We employ competing risk analyses to examine data on monitored red wolves, and model hazards and incidences for multiple causes of death plus disappearances (*endpoints* hereafter) given two types of policy interventions: (i) state policies liberalizing coyote (*C. latrans*) hunting in the North Carolina recovery area and (ii) federal issuance of red wolf 'take' permits under the ESA 10(j) rule. Coyote hunting has been a longstanding concern in red wolf management and litigation since the coyote population began to increase in the 1990s [40,44,47,48]. State coyote hunting policies are relevant because sympatric red wolves and coyotes in the North Carolina recovery area are notoriously difficult to distinguish in the field by humans and some red wolves have been mistaken for coyotes and killed (i.e. poached) [47,49]. This is particularly true during autumn hunting seasons, when red wolf pups have not attained adult-like body sizes [41,50]. Previous studies associated these hunting periods with increased anthropogenic mortality of red wolves, most notably by gunshot [37,40], and subsequent research suggests red wolf mortality is significantly higher during these periods [41]. In a prior analysis of lost-to-follow-up (LTF) red wolves, Agan *et al.* [46] found significantly more unknown fate and poached wolves than expected by chance during hunting seasons, with no significant differences for other causes of death (vehicle, non-human or legal). Hence, policy interventions liberalizing coyote hunting may have indirectly reduced protections for red wolves. As for federal policies, the USFWS directly intervened to reduce protections for red wolves for a period when they issued permits to landowners for the 'take' of red wolves ('legal' hereafter), prompting litigation and an eventual court injunction against such permits (see [51]). In other endangered US wolf populations, such reductions in federal protections were associated with increased wolf disappearances [10,11]. However, scientists have yet to directly and rigorously quantify the effect of state policy interventions or evaluate changes in hazard and incidence of red wolf mortality and disappearance in relation to policy. Our results are of importance for improving the effectiveness of policies intended to conserve endangered species and large carnivores, mitigating environmental crimes and generally improving the protection of wild animals.

# 2. Methods

## 2.1. Data collection and preparation

We analysed data acquired from the USFWS Red Wolf Recovery Program (hereafter 'Recovery Program') on radio-marked (hereafter 'collared'), monitored red wolves (i.e. within the North Carolina recovery area). The Recovery Program data include the monitoring history for all collared and monitored adult red wolves released to the wild since the beginning of releases in 1987–1 March 2020; $n = 526$. We excluded wolf pups from the dataset given most pups were marked with passive integrated transponder tags at dens, not radio collars [41]. We censored three wolves surviving past the end of our study period.

Given the small population size and limited spatial extent of the North Carolina recovery area (6000 km$^2$), the Recovery Program collared and monitored most red wolves in the wild. The Recovery Program trapped red wolves annually to fit individuals with mortality-sensitive radio collars and conducted radio-telemetry flights weekly to locate collared wolves as well as dens with pups (details in [41]). In sum, the geographic coverage and frequency of telemetry combined with on-foot surveys to locate known individuals and their pack-mates seem relatively intense compared to larger, less-intensively managed wolf populations, which helps mitigate (yet not fully eliminate) common biases occurring when marked subsamples are not representative of the entire population [52–56], such as overestimating apparent survival [18,57].

The Recovery Program survival dataset contains the following individual covariates employed in our analyses: endpoint (i.e. final wolf fate by cause of death or disappearance (LTF)), capture date (beginning of monitoring) and date of endpoint (end of monitoring). The dataset also provided data on sex, but a log-rank test on this covariate, by endpoint and globally (all endpoints), revealed no statistical differences. For recovered wolf carcasses, the cause of death was estimated by USFWS using standard methods following necropsy and radiography (for a discussion of the accuracy and precision of these methods done by USFWS in another region, see [4]). Agency removal occurred when a red wolf was captured and removed permanently (lethally or not) from the North Carolina population, generally because the wolf was considered a problem animal by USFWS but also some wolves were removed to

**Table 2.** Number of endpoints (unique wolf IDs) during periods of reduced state or federal protections (1) in the North Carolina recovery area (1987–1 March 2020). Periods of reduced protections include state policies liberalizing coyote hunting in the North Carolina recovery area as well as federal issuance of take permits under the 10(j) rule. Wolves that survived to the end of the study period are omitted here and censored at the end of the study period ($n = 3$).

| endpoint | periods of state and federal policy for red wolves (red_prot) | | total |
| --- | --- | --- | --- |
| | 0 | 1 | |
| | strict protection | reduced protection | |
| agency removal | 38 | 2 | 40 |
| collision | 56 | 12 | 68 |
| LTF | 101 | 16 | 117 |
| non-human | 60 | 6 | 66 |
| reported poached | 109 | 41 | 150 |
| unknown | 65 | 17 | 82 |
| total | 429 | 94 | 523 |
| time at risk ($t$ = days) | 467 686 | 73 410 | 541 096 |

supplement other wild populations ($n = 5$). The LTF endpoint occurred when a wolf in the wild disappeared from monitoring because the affixed radio collar stopped functioning due to either mechanical/battery failure or tampering/destruction by external causes including humans.

We reclassified the marked animals' fates obtained from the Recovery Program survival data into the following mutually exclusive endpoints, following [4,11]: agency removals (lethal or not, by agency personnel; $n = 40$, 7.6%), collision (trauma by vehicle(s); $n = 68$, 12.9%), reported poached ($n = 150$, 28.5%), non-human (unrelated to humans; e.g. disease, intraspecific strife, $n = 66$, 12.5%) and unknown (unable to discern cause of death in necropsy; $n = 82$, 15.6%). We include LTF (disappeared individuals; $n = 117$, 22.2%) as one of multiple mutually exclusive endpoints given prior work showing that censoring LTF resulted in the systematic underestimation of the hazard and incidence of poaching in other wolf populations [3,10,11]. Given poaching's illegal nature, LTF conceals 'a component of cryptic poaching, in addition to those collared individuals that moved out of radio-telemetry range or those who died from natural causes but whose radio-transmitters suffered mechanical failure beforehand' [11]. Thus, while the endpoint itself is an aggregate of various causes of disappearances, we suggest the *relative change* in LTF due to policies would be mediated largely through its cryptic poaching component. Policy-mediated variation in LTF through migration seems unlikely given the limited recovery area, intensive monitoring and clustering of LTF points among other causes of death (rather than near and beyond the perimeter of the North Carolina recovery area) and abundant vacant territory [46]. Policies affecting transmitter failures seem even less likely, and there is no suggested mechanism for such an effect [10,11,46].

We estimated the time between collaring (capture date) and endpoint in days ($t$) for each red wolf ($n = 526$) in our dataset. We calculated time collared ($t$) differently for surviving, dead, removed to captivity and LTF endpoints, following [10,11]. For our mortality endpoints, we estimated $t$ for wolves monitored by telemetry until death. For wolves relocated to captivity, we used the date of final removal to captivity by agency action. For LTF wolves, we used the last date of telemetry contact. We censored any wolves who were alive at the end of our study period (1 March 2020, $n = 3$ wolves).

We focus our analysis on periods of reduced state or federal protection for wolves, modelled as a binary variable (red_prot) identifying policy periods when (i) the North Carolina Wildlife Resources Commission (NCWRC) liberalized coyote hunting in the five-county North Carolina recovery area [37,40,45,49,51] and (ii) the USFWS Recovery Program issued red wolf 'take' permits under the ESA 10(j) rule (table 2). The NCWRC allowed daytime hunting of coyotes throughout North Carolina on 1 July 1993 (BC Daniel 2021, personal communication). Following 1 August 2012, coyote hunting subsequently went through six alternating policy periods, starting with night-time hunting on private lands, which ended on 20 November 2012 because of a preliminary injunction by a federal court to prevent red wolf mortality [48]. On 26 July 2013, the NCWRC again authorized night-time hunting of

**Table 3.** Number of events (unique wolf IDs) per endpoint and RWRP management phase (I–IV), following Hinton *et al*. [40] (1987–1 March 2020, see Methods). Wolves that survived to the end of the study period are omitted here and censored at the end of the study period ($n = 3$).

| endpoint | red wolf Recovery Program phases (*mgmt_phase*) | | | | |
| --- | --- | --- | --- | --- | --- |
| | Phase I (1) | Phase II (2) | Phase III (3) | Phase IV (4) | total |
| agency removal | 18 | 18 | 4 | 0 | 40 |
| collision | 19 | 16 | 25 | 8 | 68 |
| LTF | 21 | 37 | 53 | 6 | 117 |
| non-human | 20 | 24 | 20 | 2 | 66 |
| reported poached | 11 | 38 | 86 | 15 | 150 |
| unknown | 18 | 15 | 38 | 11 | 82 |
| total | 107 | 148 | 226 | 42 | 523 |
| time at risk ($t$ = days) | 108 440 | 157 141 | 247 607 | 27 908 | 541 096 |

coyotes in the North Carolina recovery area which ended on 12 May 2014, when a US district court again issued an injunction (*Coalition v. N.C. Wildlife Res. Comm'n*, no. 2:13-CV-60-BO). Finally, from 27 February 2015 to the present, coyote hunting was conditionally permitted in the North Carolina recovery area during the daytime.

To the periods of reduced state protections, we add a period of reduced federal ESA protections during which the agency issued permits to landowners for the take of red wolves. Since 1995, red wolves have been managed under the 10(j) rule (§10j of the ESA), which allows for the issuance of such permits. However, the USFWS did not issue such permits until 2014 and did not have to revise the 10(j) rule to do so. Thus, the agency issuance of permits reflects a change in practice rather than ESA policy. The USFWS first issued a take permit for red wolves on 6 February 2014. This agency action was then challenged in court on November 2015, culminating in a preliminary injunction preventing the take of red wolves on 28 September 2016, later made permanent on 5 November 2018 [51].

We modelled Recovery Program phases as a categorical variable (*mgmt_phase*) to control for management changes and population trends, following [40] (table 3). Phase I ('1', start of programme–federal fiscal year 1998) represents the period when the Recovery Program focused its efforts on establishing a wild red wolf population [32]. Phase II ('2', fiscal years 1999–2005) represents the period when the Recovery Program began implementing the Red Wolf Adaptive Management Plan to manage hybridization with coyotes [35]. Phase III ('3', 1 October 2006–29 June 2015) represents the period when the Recovery Program staff attempted to address state coyote management as a response to both stagnant or declining wolf population and increased rates of anthropogenic mortality [36,37,45]. Lastly, Phase IV ('4', 30 June 2015–end of analysis) represents the period during which the USFWS halted red wolf reintroductions while continuing with removals, pending an evaluation of the Recovery Program [48].

Lastly, we modelled yearly hunting period(s) as a binary variable (*hunt_period*) identifying the autumn and winter hunting periods for white-tailed deer, American black bear and waterfowl ('1', 12 September–31 January; '0' otherwise; table 4) following [46], given evidence of increased anthropogenic mortality during said periods.

Other variables are unlikely to confound the effect of policy interventions because a variable would have to co-occur in a majority of the periods of reduced protection and then change states in multiple periods of stricter protection for red wolves in addition to being widespread across the North Carolina recovery area. We searched USFWS Recovery Program documents, reports, and the scholarly literature and found no evidence of other unaccounted for human or non-human changes likely to strongly influence red wolf mortality cyclically or intermittently throughout the North Carolina recovery area (e.g. periods of disease or significant changes in environmental conditions).

## 2.2. Statistical methods

We constructed hazard and subhazard models following a competing risk framework, which is an extension of survival (time-to-event) analyses [58], following the methods in [10,11]. Survival analyses

**Table 4.** Number of events (unique wolf IDs) per endpoint by autumn/winter hunting season ('1', '0' otherwise; 1987–1 March 2020). Wolves that survived to the end of the study period are omitted here and censored at the end of the study period ($n = 3$).

| endpoint | autumn/winter hunting season (hunt_period) | | total |
| --- | --- | --- | --- |
| | 0 no hunting | 1 hunting | |
| agency removal | 22 | 18 | 40 |
| collision | 48 | 20 | 68 |
| LTF | 55 | 62 | 117 |
| non-human | 36 | 30 | 66 |
| reported poached | 45 | 105 | 150 |
| unknown | 59 | 23 | 82 |
| total | 265 | 258 | 523 |
| time at risk ($t$ = days) | 334 920 | 206 176 | 541 096 |

focus on the estimation of 'time-to-event' functions, i.e. the probability of observing a time interval ($T$), from the beginning of monitoring to an endpoint, greater than some stated value $t$, $S(t) = P(T > t)$, within a specified analysis time (our study period, 1987–1 March 2020)). Additionally, the same techniques allow for calculating the (endpoint-specific) hazard function, $h_k(t)$; the instantaneous rate of occurrence of a particular endpoint $k$ conditional on not experiencing any endpoint until that time [59–62]. Cox models estimate covariate effects on the endpoint-specific hazard as $h_k(t) = h_{0k}(t)e^{(\beta_1 x_1 + \cdots + \beta_j x_j)}$, where $h_{0k}(t)$ is an unestimated baseline hazard function (i.e. semi-parametric) and $\beta_j$ represents estimates of hazard ratios (HRs) for each covariate $x_j$ (HR < 1 indicates a reduction in hazard and HR > 1 an increase in hazard). We employed Lunn & McNeil's data augmentation method (by $k$ endpoints) along with a stratified (by endpoint) joint Cox proportional hazards model to simultaneously estimate endpoint-specific changes in HRs for each endpoint–covariate interaction [63].

However, a limitation of only relying on Cox models and hazard rates is that they do not consider competing risks. Competing risk analyses extend standard survival analyses by simultaneously considering multiple endpoints (e.g. multiple causes of death, agency removal and LTF). These analyses allow for the estimation of changes in incidence (i.e. the relative incidence) of a particular endpoint conditional on a covariate level, while accounting for the hazard of experiencing other competing endpoints. In this framework, an individual can experience the end of monitoring time by only one of multiple mutually exclusive endpoints, each associated with a particular probability (risk), so we refer to the endpoints as 'competing' over time to bring about that event.

Instead of hazards, competing risk techniques estimate the endpoint-specific cumulative incidence function (CIF), defined by the failure probability Prob($T < t$, $D = k$); that is, the *cumulative probability* of endpoint $k$ (element $D$ is an index variable that specifies *which endpoint* occurred) occurring first at time $T$ (*when* the event happened) within the study period interval defined over time $t$ in the presence of other competing endpoints [58,62,64].

Within the competing risks framework, Fine–Gray (FG) subhazard models allow for estimating differences in CIFs for a given endpoint conditional on covariates [64,65]. FG models use regression techniques similar to Cox models, except the interpretation of model parameters changes: covariate estimates are interpreted as subhazard ratios (SHRs rather than Cox's HRs), or relative incidence (instead of relative risk/hazard), in the presence of other endpoints (i.e. for each covariate $x_j$: SHR < 1 indicates a reduction in incidence, and SHR > 1 an increase in incidence). Despite both being informative and complementary, competing risk models consider more information and provide greater predictive power [62,64,66].

In this study, we exploit the complementarity of both models: our joint stratified Cox model allowed us to test the hypothesis that our management covariates affected the *rate of occurrence* (i.e. hazard) of specific endpoints, and endpoint-specific FG models allowed us to test if and how much these same covariates affected the *probability* and *incidence* of said endpoints. The endpoint-specific CIFs derived from these models allowed us to visualize any effects on incidence while considering the prevalence

of each endpoint in the population. Therefore, we used both hazard and incidence to infer the changes due to our covariates and test our hypotheses (table 1, columns A–D). As a last step, we compare CIFs derived from both models (Cox or FG) visually to assess consistency in model results and identify the most appropriate CIF model [67].

For both our joint stratified Cox proportional hazards and FG subhazard models, we assume both endpoint and time-to-endpoint for each wolf (subject) is independent of other wolves' (i.e. one wolf's monitoring history and endpoint do not inform others') and that censoring is independent of other endpoints (since we also account for LTF as an endpoint) [61], given evidence that mortality of a paired red wolf breeder is usually followed by subsequent pairing (not mortality) of the surviving breeder [40]. Since we split the monitoring history of each wolf into multiple spells to include time-dependent covariates, we cluster all analyses by a unique wolf identifier [68] to account for auto-correlation. We also evaluate compliance with the proportionality assumptions of our Cox models using Schoenfeld residuals [59–61] and control for such non-proportionality when necessary in both models through the inclusion of time-varying coefficients (tvc). Both statistical models also comply with the appropriate number of events per variable (EPV; 10 > EPV for Cox, 40–50 EPV for FG) recommended in the scientific literature to ensure accurate estimation of regression coefficients and their associated quantities for the main endpoints of interest (reported poached and LTF) (table 2) [69–71]. We selected our best Cox and FG models based on parsimony, Akaike's information criterion, Bayesian information criterion and compliance with proportionality assumptions for each model. In doing so, we avoided including covariates unless they are essential to control for. We visually assessed goodness of fit for our Cox model using a Cox–Snell residual plot [59,61]. We conducted all statistical analyses in Stata 16 (StataCorp LLC, College Station, TX, 2019).

We followed recommendations for rigorous competing risk analysis [62,64,66,72], reporting results on all endpoint-specific hazards, incidences and CIFs, synthesizing them to elucidate interactions between endpoints, but with a focus on anthropogenic causes (legal, reported poached, LTF). For example, separate analyses of Wisconsin and Mexican wolves suggested increases in both the hazard and incidence of LTF during periods of reduced ESA protections over-compensated for smaller decreases in hazard and incidence of reported wolf-poaching estimated during those same periods [10,11].

We assessed the strength of evidence for any observed changes in hazard and incidence with Bayes' factors (BF) [73], as in [10]. We used Dienes' free online BF calculator, found at http://www.lifesci.sussex.ac.uk/home/Zoltan_Dienes/inference/Bayes.htm. Dienes' BF calculator assumes parameter estimates are normally distributed with known variance, which applies well to the model coefficients obtained from Cox and FG models (for HRs and SHRs, respectively). Following the discussion by Louchouarn et al. [10] on BF specifications, we assume a half-normal function with an expected standard deviation of s.d. = point estimate [73,74]. In specifying these s.d., we follow the recommendations of [73] to select a most likely value while remaining blind to the data, as required of a registered report. We used endpoint-specific estimates from the policy treatment variable from the Wisconsin and Mexican wolf populations [10,11] to model the s.d. of the BF distributions.

We report BFs for our main endpoints of interest; i.e. reported poached and LTF. We interpreted the BF strength of evidence for each hypothesis (or null) following [73,74]: $1/3 < BF < 3$—inconclusive evidence; $BF > 3$—substantial evidence for the alternative hypothesis; $BF < 1/3$—substantial evidence for the null hypothesis (i.e. no effect).

# 3. Results

We built three stratified (by endpoint) joint Cox models (see electronic supplementary material, table S1, for all model results), as follows: Model 1 (M1) incorporates all endpoint–covariate combinations; Model 2 (M2) corrects for model violations (i.e. non-proportionality) in M1 by including a time-varying coefficient (tvc) for any covariate–endpoint interaction (**Prob** > $\chi^2$ < 0.10; see M1 diagnostics in electronic supplementary material, table S2, figures S1–S3 and figure S4 for goodness of fit); and Model 3 (M3) drops those tvcs without substantial evidence of non-proportionality in the covariate–endpoint effect (reported poached*Phase III; table 5).

We focus our analysis and discussion on the poaching-related endpoints and results from M3 given that model statistics (see electronic supplementary material, table S3) point to it as the most reliable model that complies with Cox model assumptions.

The results were largely consistent across models (see electronic supplementary material, table S1) and reveal substantial increases in the hazard of poaching-related endpoints (i.e. reported poached and

**Table 5.** HR point estimates from stratified joint Cox Model 3 (M3) for $n = 526$ adult monitored red wolves (1987–1 March 2020), including compatibility intervals (95% CI), for all covariate–endpoint interactions.
Note: $*p < 0.10$, $**p < 0.05$, $***p < 0.01$.

| covariate–endpoint | reported poached | | LTF | | agency removal | | collisions | | non-human | | unknown | |
|---|---|---|---|---|---|---|---|---|---|---|---|---|
| | HR | 95% CI | HR | 95% CI | HR | 95% CI | HR | 95% CI | HR | 95% CI | HR | 95% CI |
| reduced state/federal protections (red_prot) | 1.85*** | 1.17–2.94 | 1.05 | 0.52–2.1 | 4.22 | 0.61–29.47 | 1.24 | 0.42–3.64 | 0.92 | 0.32–2.7 | 0.87 | 0.35–2.16 |
| autumn/winter non-wolf hunting season (hunt_period) | 3.56*** | 2.45– 5.18 | 1.60** | 1.1–2.35 | 1.09 | 0.56–2.13 | 0.83 | 0.46–1.49 | 1.14 | 0.66–1.96 | 0.48*** | 0.29–0.79 |
| RWRP management phase (relative to mgmt_phase = 1) | | | | | | | | | | | | |
| Phase II | 2.38** | 1.2–4.7 | 1.20 | 0.7–2.05 | 0.73 | 0.37–1.41 | 0.64 | 0.33–1.23 | 0.03** | 0–0.45 | 0.01*** | 0–0.31 |
| Phase III | 3.06*** | 1.58–5.93 | 1.13 | 0.67–1.92 | 0.07*** | 0.02–0.29 | 0.68 | 0.37–1.28 | 0.01** | 0–0.3 | 0.99 | 0.55–1.79 |
| Phase IV | 2.97** | 1.21–7.27 | 1.11 | 0.37–3.37 | 0 | 0–0 | 2.54 | 0.66–9.77 | 0.41 | 0.07–2.29 | 3.06* | 0.95–9.85 |
| time-varying coefficients [tvc - ln(t)] | | | | | | | | | | | | |
| Phase II | — | — | — | — | — | — | — | — | 1.661** | 1.07–2.58 | 1.901** | 1.13–3.19 |
| Phase III | — | — | — | — | — | — | — | — | 1.923** | 1.03–3.57 | — | — |

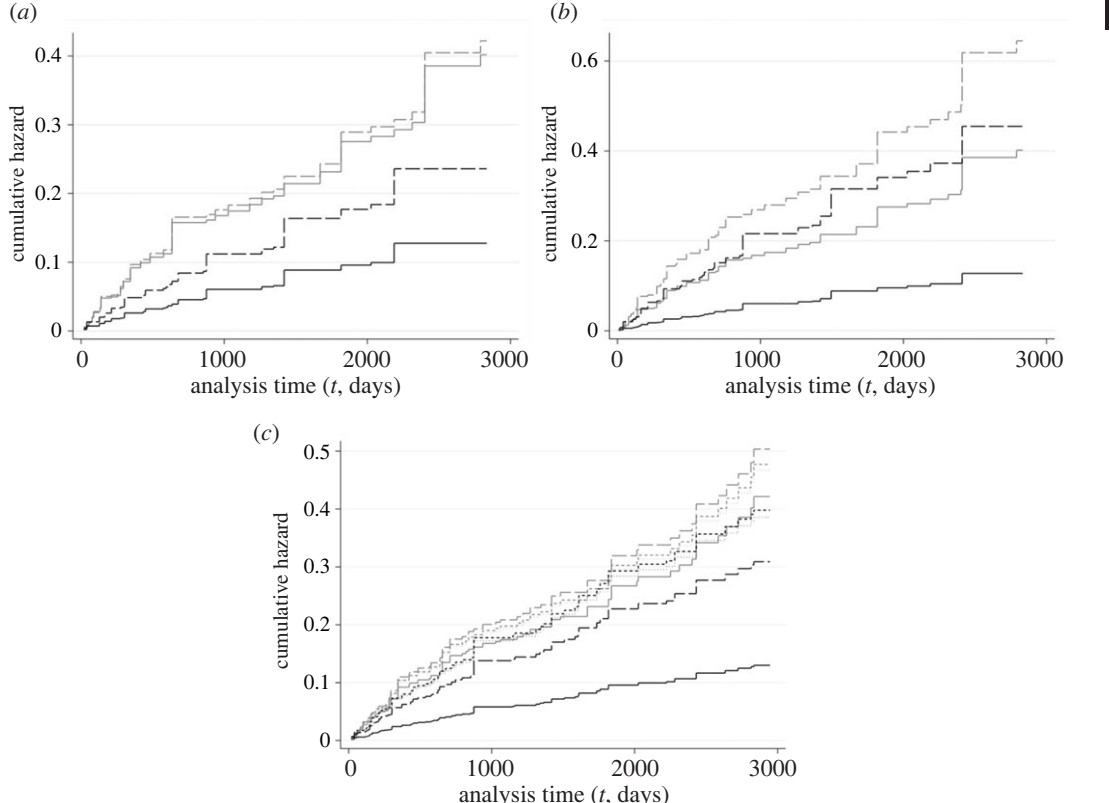

**Figure 1.** Endpoint-specific cumulative hazard curves over monitoring time (in days) for reported poached (black curves) and LTF (grey curves) derived from endpoint-season specific hazards obtained from Cox model M3 (table 1) for $n = 526$ adult monitored red wolves (1987–1 March 2020). Each panel corresponds to a covariate: ((a) *red_prot*, (b) *hunt_period*, (c) *mgmt_phase*); (a) and (b) illustrate the baseline (solid) and covariate (long-dash) cumulative hazards, whereas (c) illustrates them for RWRP management phases (Phase I: solid; Phase II: long-dash; Phase III: short-dash; and Phase IV: dot).

*LTF*; figure 1) associated with reductions in state/federal protections (including liberalizing coyote hunting) as well as yearly autumn/winter non-wolf hunting periods.

FG models for our endpoints of interest (reported poached and LTF) provide analogous results to our Cox M3 but for changes in (i.e. relative) incidence (table 6) and allow direct estimation of CIFs. We focus our analysis and discussion on FG CIFs given they follow analogous non-parametric estimates more closely than Cox-derived CIFs (from Cox M3), but results are largely consistent across models for all covariates and endpoints of interest (see electronic supplementary material, figures S5–S10). Results for all models suggest periods of reduced protections for wolves, autumn/winter non-wolf hunting seasons and management Phases II–IV were all associated with substantial increases in the incidence of both poaching-related endpoints.

## 3.1. Reported poached

*Reported poached* was the only endpoint associated with statistically significant (HRs $p < 0.10$) increases for *all* our policy and management covariates (tables 5 and 6; figures 1 and 2). Periods of reduced state and federal protections for red wolves (*red_prot*) were associated with an 85% (HR: 1.85; table 5 and figure 1*a*) increase in hazard of *reported poached*, compatible with an increase in hazard of 17–194% (95% CI), relative to periods of stricter protections. The relative incidence of *reported poached* during periods of reduced protections increased by 120% (SHR: 2.20, 95% CI: 32–266%; table 6 and figure 2*a*).

Autumn/winter non-wolf hunting seasons (*hunt_period*) were associated with increases in the hazard of *reported poached* of 256%, compatible with increases of 145–418% (95% CI), relative to the rest of the year (table 5 and figure 1*b*). Such hunting seasons are associated with increases in the relative incidence of *reported poached* of 243% (95% CI: 137–397%) relative to non-hunting periods (table 6 and figure 2*b*).

The hazard *of reported poached* also seemed to increase considerably during management Phases II–IV (*mgmt_phase*), relative to Phase I: hazard of *reported poached* increased by 138% (95% CI: +20–370%)

**Table 6.** SHR point estimates from multivariate, competing risk FG models for the reported poached (*n* = 150; 2 models) and LTF (*n* = 117) endpoints (*n* = 526 monitored red wolves), including compatibility intervals (95% CI). We built two models for reported poached to account for non-proportionality in the *mgmt_phase* covariate. Note: *p < 0.10, **p < 0.05, ***p < 0.01.

| | reported poached | | | | LTF | |
|---|---|---|---|---|---|---|
| covariate–endpoint | SHR | 95% CI | SHR | 95% CI | SHR | 95% CI |
| reduced state/federal protections (*red_prot*) | 2.31*** | 1.39–3.84 | 2.20*** | 1.32–3.66 | 1.21 | 0.58 – 2.51 |
| autumn/winter non-wolf hunting season (*hunt_period*) | 3.55*** | 2.45–5.13 | 3.43*** | 2.37–4.97 | 1.50** | 1.04–2.16 |
| RWRP management phase (relative to *mgmt_phase* = 1) | | | | | | |
| Phase II | 3.09*** | 1.55–6.17 | 0.26 | 0.03–2.57 | 1.49 | 0.86–2.6 |
| Phase III | 4.63*** | 2.39–8.96 | 0.08* | 0.01–1.18 | 1.53 | 0.9–2.6 |
| Phase IV | 4.19*** | 1.56–11.27 | 0.01 | 0–626.99 | 1.62 | 0.5–5.23 |
| time-varying coefficients [tvc - ln(*t*)] | | | | | | |
| Phase II | — | — | 1.56** | 1.02–2.38 | — | — |
| Phase III | — | — | 2.01*** | 1.26–3.22 | — | — |
| Phase IV | — | — | 2.65 | 0.55–12.74 | — | — |

during Phase II, by 206% (95% CI: +58–493%) during Phase III and by 197% (95% CI: +21–627%) during Phase IV (table 5 and figure 1*c*). Analogous FG models suggest substantial and non-proportional increases (per ln(*t*), where *t* = monitoring days) in relative incidence of reported poached during all phases relative to Phase I (table 6 tvcs, figure 2*c*), of 56% (95% CI: 2–138%) during Phase II, of 101% (95% CI: 26–222%) during Phase III and of 165% (95% CI: −55% to 1174%) during Phase IV.

## 3.2. Lost-to-follow-up

Along with *reported poached*, LTF was the only other endpoint associated with increases for all our policy and management covariates, albeit of a smaller magnitude. Reduced state and federal protections for wolves were associated with increases in the hazard of LTF by 5% relative to periods of stricter protections, with a compatibility interval ranging from decreases of 48% in hazard to increases of 110% (95% CI overlapping 0) (table 5 and figure 1*a*). Such reduced protection periods were also associated with an increase in relative incidence of LTF of 21% (95% CI: −42% to 151%; table 6 and figure 2*a*).

Autumn/winter non-wolf hunting seasons were associated with increases in the hazard of LTF of 60% (*p < 0.05*) relative to the rest of the year, compatible with a range of increases from 10 to 135% (95% CI) (table 5 and figure 1*b*). Hunting periods were also associated with increases of 50% (95% CI: 4–116%) in the relative incidence of LTF (table 6 and figure 2*b*) when compared to non-hunting periods.

Our Cox M3 model also suggests that the hazard of LTF increased during management Phases II–IV, relative to Phase I: LTF hazard increased by 20% (95% CI: −30% to +105%) during Phase II, by 13% (95% CI: −33% to +92%) during Phase III and by 11% (95% CI: −63% to +237%) during Phase IV (table 5 and figure 1*c*). Our FG model estimates higher relative increases in LTF incidence (than for hazards) for all management phases relative to Phase I: LTF incidence increased by 49% (95% CI: −14% to +160%) during Phase II, by 53% (95% CI: −10% to 160%) during Phase III and by 62% (95% CI: −50% to 423%) during Phase IV (table 6 and figure 2*c*).

## 3.3. Agency removal

Reducing federal protections for wolves was associated with increases in the hazard of agency removal (lethally or not) by 322%, with a wide set of compatible values ranging from −39% to +2847% (95% CI) given only two wolves were removed by agency staff within periods of reduced protections (table 2). Autumn/winter non-wolf hunting seasons were associated with increases in the hazard of agency removal by 9%, with a set of compatible values between −44% and +113% (95% CI). The hazard of

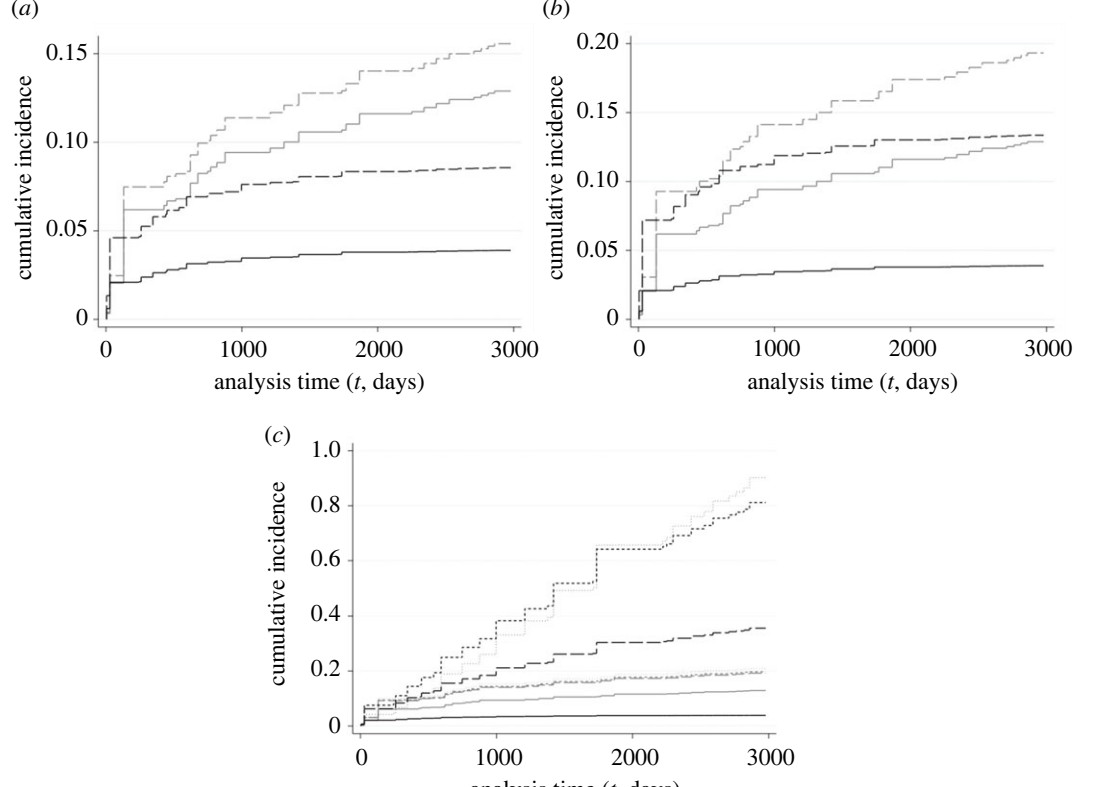

**Figure 2.** Endpoint-specific CIFs over monitoring time (in days) for reported poached (black curves) and LTF (grey curves) derived from endpoint-season specific hazards obtained from FG models (including tvcs, table 2) for $n = 526$ adult monitored red wolves (1987–1 March 2020). Each panel corresponds to a covariate: ((a) *red_prot*, (b) *hunt_period*, (c) *mgmt_phase*); (a) and (b) illustrate the baseline (solid) and covariate effect (long-dash) CIFs, whereas (c) illustrates them for RWRP management phases (Phase I: solid; Phase II: long-dash; Phase III: short-dash and Phase IV: dot).

**Table 7.** BF calculations for reported poached and LTF endpoints for adult monitored red wolves using a half-normal distribution and endpoint-specific estimates (see Methods) of HRs and SHRs for reduced protection periods (*red_prot*) from: Mexican grey wolves [from 10] and Wisconsin grey wolves [from 11] (see electronic supplementary material, table S4, for all parameters). BF strength of evidence was interpreted as follows: $1/3 < BF < 3$ would be inconclusive evidence; $BF > 3$ would represent substantial evidence for the alternative hypothesis; $BF < 1/3$ would represent substantial evidence for the null hypothesis of no association.

| | reported poached | | LTF | |
|---|---|---|---|---|
| BF specifications | HR | SHR | HR | SHR |
| Mexican grey wolves | 7.87 | 25.46 | 0.47 | 0.62 |
| Wisconsin grey wolves | 6.46 | 15.62 | 0.95 | 1.09 |

agency removal seems to have been progressively reduced relative to Phase I and through RWRP management phases: agency removal hazard decreased by 27% (95% CI −63% to +41%) during Phase II, by 93% ($p < 0.01$; 95% CI: (−) 98%–71%) during Phase III, and there were no agency removals during Phase IV.

## 3.4. Other endpoints

Periods of reduced state and federal protections were associated with increases in the hazard of collisions (24%) and decreases in the hazard of non-human (8%) and unknown (13%). Autumn/winter non-wolf hunting seasons were associated with decreases of 52% in the hazard of unknown ($p > 0.01$), a lower magnitude decrease in the hazard of collision (17%) and an increase in the hazard of non-human

(14%). RWRP management Phases II and III were associated with statistically significant ($p < 0.05$) non-proportional (tvc) increases in hazard for the non-human (Phases II and III) and unknown endpoints (Phase II), while Phase IV was associated with increases in hazard of collisions (154%) and of the unknown endpoint (206%, $p < 0.05$).

## 3.5. Bayes' factors

We calculated BFs to assess the strength of evidence for the observed changes in hazard and incidence of poaching endpoints due to reduced state or federal protections (*red_prot*), relative to endpoint-specific estimates of changes in hazard and incidence associated with reducing federal protections obtained for two other wolf populations: Mexican grey wolves (from [10]) and Wisconsin grey wolves (from [11]) (table 7; see electronic supplementary material, table S4, for all parameters). Our specification (half-normal) and use of endpoint–covariate-specific estimates test the strength of evidence for their being an increased effect relative to the effect estimated for each of the other two wolf populations.

### 3.5.1. Reported poached

All BFs suggest there is substantial evidence that the estimated increases in hazard and incidence of reported poached for collared red wolves during periods of reduced state/federal protections are higher than the increases reported for Mexican wolves and Wisconsin grey wolves (all BFs > 3). The evidence for an increased effect is stronger relative to the estimate for Mexican wolves, and more similar but still significantly different from that for Wisconsin grey wolves.

### 3.5.2. Lost-to-follow-up

All BFs suggest there is inconclusive evidence for the estimated increases in hazard and incidence of LTF for collared red wolves during periods of reduced state/federal protections being higher than those reported for the other two wolf populations (all 1/3 < BFs < 3). There is less evidence for an effect relative to the effect on Mexican wolves than for Wisconsin grey wolves.

## 4. Discussion

Our analysis of red wolf mortality replicates studies done on other US wolf populations [10,11,75] reporting significant associations between changes in hazard and incidence of poaching-related endpoints and changes in policy, management and human activities on the landscape (i.e. hunting and hounding). For monitored, adult red wolves in the wild, periods of reduced protections, autumn/winter deer and black bear hunting seasons, and Recovery Program management Phases II–IV were all associated with increases in the hazard and incidence of reported poached and disappearances (i.e. LTF). Reported poached and LTF were the only two endpoints that consistently showed increases in hazard for all covariates. Results for collared, adult red wolves reported poached were all statistically significant and substantially higher in magnitude than other endpoints, including LTF. Our results implicate intentional human behaviour (i.e. shooting rather than vehicle collisions) in mortality changes.

Overall, the reported poached endpoint was higher in the red wolf population (29%) than among Wisconsin grey wolves (17%) or Southwest US Mexican grey wolves (19%). Differences in monitoring and management may play a role in these differences. First, the red wolf monitoring programme was more intense than the other two, and monitors three taxa: red wolves, coyotes and hybrids [38,40]. Hence, it is likely that the Recovery Program was better at detecting dead animals, including red wolves. This is evidenced by the Recovery Program's management of coyotes: out of 320 sterile coyotes and hybrids monitored from 2000 to 2013, the Recovery Program lost track of 75 coyotes (24% LTF [76]; compared with 22% red wolves that ended LTF). Second, most of these radio-collared canids resided on private lands (fig. 2 in [38]), and monitoring them required coordinating with landowners, trappers and hunting clubs for access to those lands and may have contributed to a higher likelihood of locating and reporting carcasses [76]. Third, open canopy land cover such as agriculture permit long-distance (e.g. greater than 200 m) shooting of wildlife by hunters which may have resulted in poor placement of shots that caused injured animals to flee and die elsewhere (see [77]). Therefore, we speculate that many carcasses were not tampered with resulting in LTF, because poachers were unable or unwilling to locate and destroy evidence.

Our analysis focused on the effect of periods of reduced protections for wolves, which were constructed largely following policy periods when the NCWRC liberalized coyote hunting in the five-county North Carolina recovery area. By liberalizing coyote hunting in the recovery area, the NCWRC likely increased the probability that red wolves were mistaken as coyotes by hunters and killed. Our results associate such periods of liberalized coyote killing with substantial increases in the hazard of reported poaching (+85%, table 5) and minimal changes in the hazard of disappearances (+5%, table 5). Such results differ substantially from those seen in other wolf populations, where reduced protections led to increases in disappearances and fewer wolves reported poached. For Wisconsin grey wolves, reducing federal protections was associated with increases in LTF between 11 and 34%, while reported poached decreased on average 19% [11]. For Mexican grey wolves, reductions in federal protections were followed by increased LTF of 121%, while reported poached decreased by 22% [10]. The less intensive monitoring of the Mexican grey wolves and even less monitoring effort on Wisconsin grey wolves might have resulted in higher rates of tampering with collars in those regions, compared to the present study. We address intra-year covariates below.

Given that we found substantial increases in the hazard of reported poaching and minimal increases to the hazard of disappearances when protections for red wolves were loosened, our results add to the above studies in other wolf populations and further weaken the 'killing for tolerance' hypothesis (table 1, column B; summarized in [22]) argued for in federal court by the USFWS. On the contrary, our model results clearly support the hypothesis of 'facilitated illegal killing' given the substantial increases in the hazard of reported poaching of red wolves during periods of reduced state protections. However, our analysis cannot discern between poaching due to a case of mistaken identity and opportunistic poaching incentivized by devaluing red wolves. Moreover, our BF analysis provides substantial evidence that the increase in red wolves reported poached was higher than those for Mexican wolves and Wisconsin grey wolves (BFs > 3), further strengthening the evidence for the 'facilitated illegal killing' hypothesis. Despite a $HR_{ltf} > 1$, our results remain inconclusive as to the strength of evidence for the 'facilitated cryptic poaching' hypothesis, given the wide LTF 95% confidence intervals overlapping HR = 1 and inconclusive BFs. In short, we find equivocal evidence from hazard and incidence data for the hypothesis that liberalizing wolf-killing will lead to concealment of evidence of poaching rather than just leading to more poaching. We return to the question below when discussing monitoring efforts and methods.

This is the third study of a US wolf population that provides support for the 'facilitated illegal killing' hypothesis and evidence against the 'tolerance killing' hypothesis promoted by state and federal agencies [10,11]. The implication is that policies that liberalize legal wolf-killing at the federal or state level also increase illegal killing, the latter at a much higher rate. On the contrary, policies that restrict killing and increase protections for wolves mitigated their intentional killing, both legal and illegal. Thus, despite poaching potentially being considered a form of resistance to either increased protections or growing wolf numbers, decreasing protections is not a credible policy solution, but rather exacerbates the problem, perhaps by devaluing wolves publicly.

Of all our covariates, the autumn/winter deer and black bear hunting seasons we examined also coincided with the highest increases in hazard of poaching-related endpoints for red wolves: greater than 250% increase for reported poached and a 60% increase in LTF during said periods. Our results suggest hunting seasons may be the most important factor (more than state and federal policies or management) mediating poaching risk for red wolves. Agan *et al.* [78] conducted interviews and a survey of attitudes towards red wolves within the North Carolina recovery area and found majority positive public attitudes towards red wolves and disinclination to poach, adding that 'respondents with positive attitudes toward red wolves had lower inclinations to poach them' [78, p. 4]. That study also found extremely low inclinations to poach (median of 0–1 out of possible 0–9 scale; higher numbers indicating higher inclinations to poach) when compared to an analogous study conducted in Wisconsin [27]. However, for both red wolves and Wisconsin grey wolves, autumn and winter hunting seasons seem to be the most significant factor mediating both poaching-related endpoints (reported poached and LTF). Indeed, for Wisconsin grey wolves, autumn and winter deer and black bear hunting seasons during periods with snow cover (which may allow for increased detection of wolves) substantially and significantly increase the hazard for grey wolves reported poached (by >650% relative to seasons without said factors), with lower increases (of 19%) in LTF [75]. Results for both populations suggest that (i) environmental conditions such as open canopy cover characteristics of agricultural landscapes expose red wolves to poaching in the North Carolina recovery area), (ii) the surge of hunters on the landscape during the hunting seasons provides cover for illegal activity (increase in overall poaching), (iii) long-distance shooting opportunities decrease the

inclination of poachers to track down their illegal kills and destroy radio collars and (iv) intensive monitoring of red wolves and coyotes by a federal agency increases the detection of poaching in the North Carolina recovery area (higher increase in reported poached than LTF). Moreover, poaching seems to be carried out by a minority of individuals, with hunters reporting the highest inclinations to do so [78]. The substantial increase in disappearances during hunting periods also points to the importance of cryptic poaching to total mortality and how dismissing disappeared individuals, or assuming they die at the same rates and causes as those wolves that are found, will certainly underestimate mortality, misidentify the causes of death that deserve intervention, and overestimate population sizes [3,10,11,75]. Because both reported poached and LTF are the most prevalent endpoints in both populations, we surmise seasons of anthropogenic activity that either invite the presence of or provide cover for individuals with inclinations to poach are more important than climatic seasons for overall wolf mortality [75].

Our models also associated Recovery Program management Phases II–IV with substantial and significant increases in the hazard of red wolves reported poached (from 138% in Phase II to around 300% for Phases III–IV, relative to Phase I), as well as lower increases in the hazard of LTF (from 20% in Phase II to 11% in Phase IV). Most management phases (Phases I–III) mainly represent on-the-ground responses by the federal wildlife agency USFWS to coyote colonization and hybridization. The exception being Phase IV, when the USFWS aligned its management with state wildlife agency NCWRC demands, halted red wolf reintroductions, and the population collapsed from 113–149 (2013–2014) to 74 (2014–2015) [48,79]. The estimated increases in poaching during Phases II–III might be argued to relate to red wolf population growth and coyote colonization of the North Carolina recovery area [76]; i.e. individuals increased their rate of poaching given the higher abundance of both canids, potentially mistaking some wolves for coyotes. Notably, non-human mortalities also increased significantly and non-proportionally during Phases II–III relative to Phase I. By the end of Phase III, these increases in hazard had caused the red wolf population to decline to levels similar to those of Phase I [41,80,81]. By Phase IV, the red wolf population had plummeted to 74 individuals or less, yet poaching hazard remained high (reported poached increased by almost 200%). Moreover, there were also substantial increases in other endpoints during Phase IV: red wolf deaths by vehicle collision increased by 154%, while deaths by unknown causes increased by greater than 200%. Our results suggest that, despite consistent decreases in the hazard of agency removals, policy changes associated with management were unable to mitigate poaching or other types of anthropogenic mortality. For red wolves, policies liberalizing coyote killing in the North Carolina recovery area potentially discouraged hunter precaution in selecting a target. Arguably, exposing red wolves to shooting deaths through liberalization of coyote killing, even as the red wolf population declines, conveys their devaluation by state and federal agencies.

To mitigate red wolf-killing, the state could ban coyote killing within the North Carolina recovery area (as done by court injunction between July 2013 and May 2014; *Red Wolf Coalition v. N.C. Wildlife Res. Comm'n*, No. 2:13-CV-60-BO) while ramping up efforts to protect red wolves and coyotes from poaching via increased use of law enforcement. Such policy measures seem achievable through the ESA Sec. 4(e) (Similarity of Appearance Cases) and seem not only necessary but urgent given that poaching is the main cause of mortality for red wolves and an impediment to population growth and recolonization. Our study indicates the protection of both coyotes and red wolves in the North Carolina recovery area may be an effective way to reduce anthropogenic mortality, which will allow for the establishment of a healthy red wolf population and minimize hybridization [39,40]. Therefore, protecting both canids in the North Carolina recovery area would align with the ESA and red wolf recovery plans, making the strategy resistant to court challenge. Going further, given the perilous state of the red wolf population in the North Carolina recovery area and the mandate to protect and recover it, restricting hunting while increasing anti-poaching interventions should be an urgent priority.

The scientific consensus suggests the USFWS should reconsider the red wolf's designation as a 10(j) 'experimental non-essential' population. 'Experimental essential' and 'non-essential' populations are treated as 'threatened' species under the ESA, which allows incidental and unintentional take as well as the intentional take by landowners in response to nuisance or conflicts. Relative to stricter levels of protections, i.e. 'endangered', the 'experimental' and even more so the 'non-essential' designation communicates a lower value of the individuals of these struggling, small populations (which includes no designation of 'critical habitat'). Our results suggest that the USFWS is making its own job of recovering endangered populations more difficult. It could save time and attendant taxpayer money and legal challenges which the USFWS has generally not won (see *Ctr. for Biological Diversity v. Jewell*,

No. CV-15–00019-TUC-JGZ (l) (D. Ariz. 30 March 2018) and *Red Wolf Coal. v. U.S. Fish & Wildlife Serv.*, no. 2:20-CV-75-BO (E.D.N.C. 21 January 2021)). Signalling the increased value of red wolves in federal and state policy would align with widespread support for red wolf recovery inside the North Carolina recovery area and beyond [78]. Policies liberalizing lethal methods, allowed even under a 10(j) 'essential experimental' population designation, seem to incentivize illegal killing and concealment by a minority of individuals, which runs contrary to conservation goals and agency public trust responsibilities [2]. We recommend instead that wildlife agencies attempting to recover listed species keep strict protections until populations achieve geographic and numerical invulnerability to poaching judged by the best-available science.

Data accessibility. The pre-registered Stage 1 report can be found on the Open Science Framework at the following link: https://osf.io/wk6ay/?view_only=d3649ffc01834bcfabfbb2fa8017fd51. The raw dataset from the USFWS Red Wolf Recovery Program has been submitted to Dryad and can be found at the following link: https://doi.org/10.5061/dryad.8cz8w9gsr [82]. We have also included the prepared and processes data in our Dryad submission, which is ready to be run through the STATA code included in the electronic supplemental material.

The data are provided in the electronic supplementary material [83].

Authors' contributions. F.J.S.-Á.: conceptualization, data curation, formal analysis, investigation, methodology, project administration, resources, software, validation, visualization, writing—original draft and writing—review and editing; S.A.: conceptualization, data curation, investigation, resources, validation and writing—review and editing; J.W.H.: data curation, investigation, resources, validation and writing—review and editing; A.T.: conceptualization, funding acquisition, validation and writing—review and editing.

All authors gave final approval for publication and agreed to be held accountable for the work performed therein.

Conflict of interest declaration. F.J.S.-Á., S.A. and J.W.H. declare no competing interests. A.T. declares no competing interests and provides his CV (http://faculty.nelson.wisc.edu/treves/archive_BAS/Treves_vita_Jan2020.pdf) and all funding awarded as of 6 January 2020 (http://faculty.nelson.wisc.edu/treves/archive_BAS/funding.pdf) for transparency, so readers can decide if they perceive a competing interest.

Funding. Funding for F.J.S.-Á. salary was provided by the University of Wisconsin-Madison WARF (Wisconsin Alumni Research Foundation) as a Vilas Lifecycle grant to A.T.

Acknowledgements. We thank the staff at the US Fish & Wildlife Service Red Wolf Recovery Program for data collection, provision and assistance in data interpretation. We thank the UCLA Law School Animal Law and Policy Grants Program and Therese Foundation for funding. We thank the editor and two anonymous reviewers for their helpful comments and suggestions. This article does not necessarily reflect the views of the institutions or agencies involved.

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
