## [Peer Review File · Royal Society Open Science]

Review History

RSOS-210400.R0 (Original submission)

Review form: Reviewer 1

Do you have any ethical concerns with this paper?

No

Recommendation?

Accept with minor revision

Comments to the Author(s)

The authors have presented the background and methods for a proposed study evaluating how management polices affect red wolf mortality and disappearance.

It has been submitted as a registered report - stage 1. I have therefore focused my review under the suggested headings provided.

Line numbers are added for specific comments.

Scientific validity:

This is an important study that uses a long term data set on red wolf survival. The data set offers a unique opportunity to assess how different management strategies impacted endangered red wolves.

The logic, rationale, and plausibility of the proposed hypotheses:

The main hypothesis tested is that interventions in federal or state policies that exposed red wolves to more anthropogenic killing without explicitly loosening the endangered species protections would suffice to promote poaching. This is a valid hypothesis and there is good justification provided to propose and explore it. More specific details are provided in table 4, however, and I wonder if this level of detail should be referenced earlier in the text, as they are more specific hypotheses than what is listed in the introduction?

The soundness and feasibility of the methodology and analysis pipeline (including statistical power analysis where applicable):

I have no concerns about the dataset being used. I do, however, have some comments in relation to methods text and the proposed analyses.

line 185: provide more details about the differences between marked and monitored wolves - any differences in the levels of certainty about end points?

line 188: provide more justification for excluding wolf pups.

lines 201-212: it would help to provide some more details up front here about how you incorporated lost-to-follow-up (LTF) covariates in the analysis, along with more better justification of using this covariate. You state that LTF data relate to incidence when the wolves disappeared from monitoring, but in lines 219-221 you state that you treat this data as one of multiple mutually-exclusive endpoints. Does this mean you treat this data in the same way as a mortality? 22% of the data relates to LTFs and you mention that there has been systematic under-estimation of poaching; but does treating this data essentially as a mortality overestimate incidences of hazards and poaching? Better justification is required here.

line 234: The main hypothesis mentions "more anthropogenic killing without explicitly loosening the endangered species protections" but here you mention a period where red wolf 'take' permits are issued under the ESA 10(j) - isn't this an instance where protections are loosened at a federal level? In addition, your main hypothesis relates to poaching, but at this point in the analysis it seems the focus is on all forms of mortality? Perhaps some more context would help. I note, however, that Table 4 is a good addition/summary of the more specific hypotheses.

Lines 231-263: It took a long time to link the dates in lines 231-250, with the phases in lines 252-263. Suggest to merge together as I can't follow the logic of the dates.

lines 282-350: The survival analysis approach is sound but there is not sufficient detail provided about all the assumptions required for this sort of test to function properly e.g. independence between samples, crossing hazard or survival functions etc. I would add something more in here, as you only focus on having appropriate events per variable.

line 333: I can see here that you separate the end points - this is valid, and should be mentioned earlier so there is no confusion about lumping all the data together.

Whether the clarity and degree of methodological detail would be sufficient to replicate exactly the proposed experimental procedures and analysis pipeline:

Yes, the detail is good, pending a few issues noted above.

Whether the authors provide a sufficiently clear and detailed description of the methods to prevent undisclosed flexibility in the experimental procedures or analysis pipeline:

Yes.

Whether the authors have considered sufficient outcome-neutral conditions (e.g. positive controls) for ensuring that the results obtained are able to test the stated hypotheses:

Table 4 covers this well.

Minor comments:

Line 79: spell out WI and same for MI (there are people outside the US who will read this paper).

Introduction: general comment that it is very long and could be shortened.

Review form: Reviewer 2

Do you have any ethical concerns with this paper?

No

Recommendation?

Accept in principle

Comments to the Author(s)

Many congratulations for the useful studies that would be helpful in conserving the species for long term. I suggest it to be published. There are some minor comments and some questions to be considered...

Decision letter (RSOS-210400.R0)

Dear Dr Santiago-Ávila,

On behalf of the Editors, I am pleased to inform you that your Manuscript RSOS-210400 entitled "Evaluating how management policies affect red wolf mortality and disappearance" deemed suitable for in-principle acceptance in Royal Society Open Science subject to minor revision in accordance with the referee and editor suggestions. Please find their comments at the end of this email.

The reviewers and handling editors have recommended publication, but also suggest some minor revisions to your manuscript. Therefore, I invite you to respond to the comments and revise your manuscript.

Please you submit the revised version of your manuscript within 30 days (i.e. by the 03-Jun-2021). If you do not think you will be able to meet this date please let me know immediately.

Full author guidelines can be found here <https://royalsocietypublishing.org/rsos/registered-reports#ReviewerGuideRegRep>.

Kind regards
Professor Chris Chambers
Royal Society Open Science
openscience@royalsociety.org

on behalf of Professor Chris Chambers
(Subject Editor, Royal Society Open Science)
openscience@royalsociety.org

Associate Editor Comments to Author (Professor Chris Chambers):

Thanks for your patience during this challenging time for reviewers. Two expert reviewers have now assessed the manuscript, and the good news is that their assessments are broadly very positive and bode well for achieving Stage 1 in-principle acceptance (IPA). As you will see, the reviewers are enthusiastic about the motivation for the study, with comments mainly prompting the inclusion of greater methodological detail and specific clarifications, as well as stronger justification for design decisions and ensuring close alignment between the hypotheses and analysis plans. Provided the authors are able to respond comprehensively to these points in a revised submission, IPA should be forthcoming without requiring further in-depth Stage 1 review.

Reviewer comments to Author:

Reviewer: 1

Comments to the Author(s)

The authors have presented the background and methods for a proposed study evaluating how management polices affect red wolf mortality and disappearance.

It has been submitted as a registered report - stage 1. I have therefore focused my review under the suggested headings provided.

Line numbers are added for specific comments.

Scientific validity:

This is an important study that uses a long term data set on red wolf survival. The data set offers a unique opportunity to assess how different management strategies impacted endangered red wolves.

The logic, rationale, and plausibility of the proposed hypotheses:

The main hypothesis tested is that interventions in federal or state policies that exposed red wolves to more anthropogenic killing without explicitly loosening the endangered species protections would suffice to promote poaching. This is a valid hypothesis and there is good justification provided to propose and explore it. More specific details are provided in table 4, however, and I wonder if this level of detail should be referenced earlier in the text, as they are more specific hypotheses than what is listed in the introduction?

The soundness and feasibility of the methodology and analysis pipeline (including statistical power analysis where applicable):

I have no concerns about the dataset being used. I do, however, have some comments in relation to methods text and the proposed analyses.

line 185: provide more details about the differences between marked and monitored wolves - any differences in the levels of certainty about end points?

line 188: provide more justification for excluding wolf pups.

lines 201-212: it would help to provide some more details up front here about how you incorporated lost-to-follow-up (LTF) covariates in the analysis, along with more better justification of using this covariate. You state that LTF data relate to incidence when the wolves disappeared from monitoring, but in lines 219-221 you state that you treat this data as one of multiple mutually-exclusive endpoints. Does this mean you treat this data in the same way as a mortality? 22% of the data relates to LTFs and you mention that there has been systematic under-estimation of poaching; but does treating this data essentially as a mortality overestimate incidences of hazards and poaching? Better justification is required here.

line 234: The main hypothesis mentions "more anthropogenic killing without explicitly loosening the endangered species protections" but here you mention a period where red wolf 'take' permits are issued under the ESA 10(j) - isn't this an instance where protections are loosened at a federal level? In addition, your main hypothesis relates to poaching, but at this point in the analysis it seems the focus is on all forms of mortality? Perhaps some more context would help. I note, however, that Table 4 is a good addition/summary of the more specific hypotheses.

Lines 231-263: It took a long time to link the dates in lines 231-250, with the phases in lines 252-263. Suggest to merge together as I can't follow the logic of the dates.

lines 282-350: The survival analysis approach is sound but there is not sufficient detail provided about all the assumptions required for this sort of test to function properly e.g. independence

between samples, crossing hazard or survival functions etc. I would add something more in here, as you only focus on having appropriate events per variable.

line 333: I can see here that you separate the end points - this is valid, and should be mentioned earlier so there is no confusion about lumping all the data together.

Whether the clarity and degree of methodological detail would be sufficient to replicate exactly the proposed experimental procedures and analysis pipeline:

Yes, the detail is good, pending a few issues noted above.

Whether the authors provide a sufficiently clear and detailed description of the methods to prevent undisclosed flexibility in the experimental procedures or analysis pipeline:

Yes.

Whether the authors have considered sufficient outcome-neutral conditions (e.g. positive controls) for ensuring that the results obtained are able to test the stated hypotheses:

Table 4 covers this well.

Minor comments:

Line 79: spell out WI and same for MI (there are people outside the US who will read this paper).

Introduction: general comment that it is very long and could be shortened.

Reviewer: 2

Comments to the Author(s)

Many congratulations for the useful studies that would be helpful in conserving the species for long term. I suggest it to be published. Below are some minor comments and some questions:

1. It would be interesting to include the cultural factors that affects poaching practices in the Introduction.
2. I was wondering if societal responses to poaching was looked at, if yes, should be included in the Introduction.
3. Have authors looked at any impact of animal poaching on wildlife tourism...?
4. If yes, does a tourist boycott due to local poaching threat?
5. How effective are laws on wildlife crime, for example The Department of Justice's Environment and Natural Resources Division (ENRD), together with United States Attorneys are responsible for prosecuting international wildlife trafficking crimes, primarily under the Endangered Species Act (ESA) and the Lacey Act.
6. Is there any impact on international laws on wildlife conservation?
7. Does harvests made without complying with the regulations for legal harvest result in the illegal taking of wildlife and come under poaching, where only wildlife can be poached.
8. As it happens elsewhere, was there any impact of poverty that led to animals poaching?
9. It would be interesting to investigate number of animals are poached annually?
10. The current wave of poaching is carried out by sophisticated and well-organized criminal networks - using helicopters, night-vision equipment, tranquilizers and silencers to kill animals at night, avoiding law enforcement patrols. Are the laws effective to curtail these aspects to protect carnivores and prey bases?
11. Line 10: illegal killing (poaching) should be one of the causes for deaths of carnivores
12. Use 'Threatened species' rather endangered (IUCN standard term) line26

13. One or two lines should be included on result output in the Abstract.
14. Line 43: I suggest poaching is one of the many causes
15. Line: 50 use threatened instead of endangered
16. Line 91-96: how legally killing associated with fewer disappearances?
17. Line 119-1122: It is little contradictory when Mexican gray wolves' deaths and disappearances un controlled condition and reduced federal protections and higher disappearances.
18. Line147-158: if hybridization between coyotes and wolves defined as one the major threats, was the mitigation was effective? While the captures from the wild extirpated the species in 1996, and further gunshot deaths of red wolves in later years put pressure in later years. Does the Reintroduction was successful?
19. Line164-180: when it is difficult to distinguish coyotes and red wolves and constantly killing either of these, how the reintroduction program effective? It would be interesting to look at this aspect.
20. Line 200-212: Bases on various studies, many times, the collared individuals persist in suitable areas, while the radio collars stop working, wild population may vary and disappearances of species as well...
21. Line 214-221: It is interesting to note that poaching is 28.3% and disappeared individuals was 22%, when considered point 20, it would result differently.
22. Line 236-250: based on point 19, when coyotes were hunted day and night and people might be taking down red wolves, how agencies protect the species.
23. Table 4 the analysis and interpretation on hypotheses are rightly depicted
24. I think it is a great studies and need to be published.

Author's Response to Decision Letter for (RSOS-210400.R0)

See Appendix A.

Decision letter (RSOS-210400.R1)

Dear Dr Santiago-Ávila

On behalf of the Editor, I am pleased to inform you that your Stage 1 Registered Report RSOS-210400.R1 entitled "Evaluating how management policies affect red wolf mortality and disappearance" has been accepted in principle for publication in Royal Society Open Science.

You may now progress to Stage 2 and complete the study as approved. Before commencing data collection we ask that you:

- 1) Update the journal office as to the anticipated completion date of your study.
- 2) Register your approved protocol on the Open Science Framework (<https://osf.io/>) or other recognised repository, either publicly or privately under embargo until submission of the Stage 2 manuscript. Please note that a time-stamped, independent registration of the protocol is mandatory under journal policy, and manuscripts that do not conform to this requirement cannot

be considered at Stage 2. The protocol should be registered unchanged from its current approved state, with the time-stamp preceding implementation of the approved study design. We recommend using the dedicated registration process for accepted Registered Reports at <https://osf.io/rr> to register your submission.

Following completion of your study, we invite you to resubmit your paper for peer review as a Stage 2 Registered Report. Please note that your manuscript can still be rejected for publication at Stage 2 if the Editors consider any of the following conditions to be met:

- The results were unable to test the authors' proposed hypotheses by failing to meet the approved outcome-neutral criteria.
- The authors altered the Introduction, rationale, or hypotheses, as approved in the Stage 1 submission.
- The authors failed to adhere closely to the registered experimental procedures. Please note that any deviations from the approved experimental procedures must be communicated to the editor immediately for approval, and prior to the completion of data collection. Failure to do so can result in revocation of in-principle acceptance and rejection at Stage 2 (see complete guidelines for further information).
- Any post-hoc (unregistered) analyses were either unjustified, insufficiently caveated, or overly dominant in shaping the authors' conclusions.
- The authors' conclusions were not justified given the data obtained.

We encourage you to read the complete guidelines for authors concerning Stage 2 submissions at <https://royalsocietypublishing.org/rsos/registered-reports#ReviewerGuideRegRep>. Please especially note the requirements for data sharing, reporting the URL of the independently registered protocol, and that withdrawing your manuscript will result in publication of a Withdrawn Registration.

Once again, thank you for submitting your manuscript to Royal Society Open Science and we look forward to receiving your Stage 2 submission. If you have any questions at all, please do not hesitate to get in touch. We look forward to hearing from you shortly with the anticipated submission date for your stage two manuscript.

on behalf of Professor Chris Chambers (Registered Reports Editor, Royal Society Open Science)
openscience@royalsociety.org

Author's Response to Decision Letter for (RSOS-210400.R1)

See Appendix B.

RSOS-210400.R2

Review form: Reviewer 2

Is the manuscript scientifically sound in its present form?

Yes

Are the interpretations and conclusions justified by the results?

Yes

Is the language acceptable?

Yes

Do you have any ethical concerns with this paper?

No

Have you any concerns about statistical analyses in this paper?

No

Recommendation?

Accept as is

Comments to the Author(s)

Many congratulations on the great work. Hope to see more similar work in the future. All the best(see Appendix C).

Decision letter (RSOS-210400.R2)

Dear Dr Santiago-Ávila:

It is a pleasure to accept your manuscript entitled "Evaluating how management policies affect red wolf mortality and disappearance" in its current form for publication in Royal Society Open Science. The comments of the reviewer(s) who reviewed your manuscript are included at the foot of this letter.

Thank you for your fine contribution. On behalf of the Editors of Royal Society Open Science, we look forward to your continued contributions to the journal.

on behalf of Professor Chris Chambers (Subject Editor)
openscience@royalsociety.org

Associate Editor Comments to Author (Professor Chris Chambers):
Comments to the Author:

One of the original Stage 1 reviewers was available to evaluate the Stage 2 manuscript, and is satisfied with the submission in its current state. I have also read the manuscript and find that it meets the Stage 2 criteria, so we are in the unusual (and happy) situation of being able to award acceptance without revision.

One technical point, which can be resolved at the proof stage: please ensure that the OSF project containing the approved Stage 1 manuscript is made public and that the private view-only link to the project (https://osf.io/wk6ay/?view_only=d3649ffc01834bcfabfbb2fa8017fd51) is replaced with the accessible public link: <https://osf.io/wk6ay/>
Please note that the Dryad data repository also needs to be made fully public.

In the meantime, congratulations on a very well executed Registered Report.

Reviewer comments to Author:

Reviewer: 2

Comments to the Author(s)

Many congratulations on the great work. Hope to see more similar work in the future. All the best.

Appendix A

Response to Reviewers

Associate Editor Comments to Author (Professor Chris Chambers):

Thanks for your patience during this challenging time for reviewers. Two expert reviewers have now assessed the manuscript, and the good news is that their assessments are broadly very positive and bode well for achieving Stage 1 in-principle acceptance (IPA). As you will see, the reviewers are enthusiastic about the motivation for the study, with comments mainly prompting the inclusion of greater methodological detail and specific clarifications, as well as stronger justification for design decisions and ensuring close alignment between the hypotheses and analysis plans. Provided the authors are able to respond comprehensively to these points in a revised submission, IPA should be forthcoming without requiring further in-depth Stage 1 review.

RESPONSE(s): Thank you for the constructive feedback, suggestions and the opportunity to revise and response. We believe we have successfully addressed or responded to all reviewer queries below and in our manuscript. The line numbers in our responses below refer to the 'Main Document' (doc name "...CLEAN_FINAL"), rather than the submitted 'Review File' reflecting tracked changes.

Reviewer comments to Author:

Reviewer: 1

Comments to the Author(s)

The authors have presented the background and methods for a proposed study evaluating how management polices affect red wolf mortality and disappearance.

It has been submitted as a registered report - stage 1. I have therefore focused my review under the suggested headings provided.

Line numbers are added for specific comments.

Scientific validity:

This is an important study that uses a long term data set on red wolf survival. The data set offers a unique opportunity to assess how different management strategies impacted endangered red wolves.

The logic, rationale, and plausibility of the proposed hypotheses:

The main hypothesis tested is that interventions in federal or state policies that exposed red wolves to more anthropogenic killing without explicitly loosening the endangered species protections would suffice to promote poaching. This is a valid hypothesis and there is good justification provided to propose and explore it. More specific details are

provided in table 4, however, and I wonder if this level of detail should be referenced earlier in the text, as they are more specific hypotheses than what is listed in the introduction?

We have attempted to clarify by moving Table 4 to Table 1 and referencing its information much earlier in the text, as in both we describe the 3 hypotheses by name. That is, without addressing the statistical details (Analysis Plan and Interpretation... in Table 1, columns C-D and pp. 5-6).

The soundness and feasibility of the methodology and analysis pipeline (including statistical power analysis where applicable):

I have no concerns about the dataset being used. I do, however, have some comments in relation to methods text and the proposed analyses.

line 185: provide more details about the differences between marked and monitored wolves - any differences in the levels of certainty about end points?

To clarify, marked, monitored wolves are only one classification (not two, as may have been interpreted by the reviewer). As stated in the Methods, “For recovered wolf carcasses, cause of death was estimated by USFWS using standard methods following necropsy and radiography [for a discussion of the accuracy and precision of these methods done by USFWS in another region, see [4].” Hence, we are confident that we have a replicable and independent classification of endpoint for all wolves in the dataset. Although a few errors were reported for another dataset using similar methods [4], the errors were fewer than 2% and the bias if any should not relate to policy.

line 188: provide more justification for excluding wolf pups.

Thank you for pointing this out. Previously we erroneously stated that some wolf pups were monitored. However, that is not the case, and we have clarified accordingly in lines 194-196: “We excluded wolf pups from the dataset given most pups were marked with passive integrated transponder (PIT) tags at dens and not monitored with radio collars [37].”

lines 201-212: it would help to provide some more details up front here about how you incorporated lost-to-follow-up (LTF) covariates in the analysis, along with more better justification of using this covariate. You state that LTF data relate to incidence when the wolves disappeared from monitoring, but in lines 219-221 you state that you treat this data as one of multiple mutually-exclusive endpoints. Does this mean you treat this data in the same way as a mortality? 22% of the data relates to LTFs and you mention that there has been systematic under-estimation of poaching; but does treating this data essentially as a mortality overestimate incidences of hazards and poaching? Better justification is required here.

We refer to disappearances as an endpoint throughout the manuscript starting in line 162, never as a covariate, so we perceive no discrepancy. LTF is indeed classified as one mutually-exclusive endpoint, along with every other mortality (see lines 207-238). We have addressed the reviewer's concern regarding overestimation of poaching by including LTF in lines 228-238.

line 234: The main hypothesis mentions "more anthropogenic killing without explicitly loosening the endangered species protections" but here you mention a period where red wolf 'take' permits are issued under the ESA 10(j) - isn't this an instance where protections are loosened at a federal level? In addition, your main hypothesis relates to poaching, but at this point in the analysis it seems the focus is on all forms of mortality? Perhaps some more context would help. I note, however, that Table 4 is a good addition/summary of the more specific hypotheses.

Thank you for pointing this out and providing the opportunity to clarify. We have made the relevant in-text clarifications in lines 265-267, indicating that the issuance of permits did not involve any change in the relevant policy (they were always able to issue permits under the 1995 10(j) rule), only a change in practice by the agency (they decided to start issuing them). We hope this clarifies.

Lines 231-263: It took a long time to link the dates in lines 231-250, with the phases in lines 252-263. Suggest to merge together as I can't follow the logic of the dates.

We split the relevant paragraphs by variable, explaining the categories and dates relevant to our reduced protection policy periods in lines 248-271, and recovery program management phases in 273-284. It is our impression that the categories and relevant dates for each variable are best presented and explained in this way, rather than merging categories and dates relevant to either. Following the reviewer's comment, we have edited and streamlined the relevant text hoping readers can visualize the policy changes more clearly.

lines 282-350: The survival analysis approach is sound but there is not sufficient detail provided about all the assumptions required for this sort of test to function properly e.g. independence between samples, crossing hazard or survival functions etc. I would add something more in here, as you only focus on having appropriate events per variable.

Thank you for pointing this out. We have included model details (lines 311-314), and assumptions (independence of subjects and censoring), including addressing potential non-proportionality of hazards/incidences, and final CIF selection (between Cox and FG CIFs), within the paragraphs in lines 349 and 366.

line 333: I can see here that you separate the end points - this is valid, and should be mentioned earlier so there is no confusion about lumping all the data together.

Thank you for pointing this out. This observation allowed us to further clarify our use of a joint stratified Cox proportional hazards model that estimates all

endpoint-specific hazards simultaneously, following Lunn & McNeil (1995). We have clarified this in lines 311-314. This technique still provides endpoint-covariate-specific estimates, which we mention in the initial paragraph detailing our statistical methods (line 300).

Whether the clarity and degree of methodological detail would be sufficient to replicate exactly the proposed experimental procedures and analysis pipeline:

Yes, the detail is good, pending a few issues noted above.

Whether the authors provide a sufficiently clear and detailed description of the methods to prevent undisclosed flexibility in the experimental procedures or analysis pipeline:

Yes.

Whether the authors have considered sufficient outcome-neutral conditions (e.g. positive controls) for ensuring that the results obtained are able to test the stated hypotheses:

Table 4 covers this well.

Minor comments:

Line 79: spell out WI and same for MI (there are people outside the US who will read this paper).

Thank you. We have changed all first instances of state abbreviations to proper names.

Introduction: general comment that it is very long and could be shortened.

We understand the comment and attempted to shorten, but found everything but a couple of sentences were integral to the Background.

Reviewer: 2

Comments to the Author(s)

Many congratulations for the useful studies that would be helpful in conserving the species for long term. I suggest it to be published. Below are some minor comments and some questions:

1. It would be interesting to include the cultural factors that affects poaching practices in the Introduction.

We agree it would be interesting. However, we consider the introduction is already full of background (see R1's comment that it is already long) that seems more relevant to our analysis. We look forward to addressing reviewer 2's interest in our discussion.

2. I was wondering if societal responses to poaching was looked at, if yes, should be included in the Introduction.

In our future Discussion, we will address the broad literature on public attitudes to poaching and cite recent work by Agan on attitudes to red wolves and their conservation.

3. Have authors looked at any impact of animal poaching on wildlife tourism...?
In our Discussion, we will address the long-held hypothesis that tourists and researchers can suppress poaching in places like Africa. That may well be the case in the USA too –Also please see Santiago-Ávila et al. 2020 which hinted that a certain period of wolf census in Wisconsin that was characterized by large numbers of private citizens counting wolves was associated with a decline in reported poaching. Although that topic is interesting, we feel it is not sufficiently relevant to red wolves at present because the small number of animals, their heavy use of private lands, and the absence of large, strictly protected areas (Agan et al. 2021) seem to make tourism a marginal, non-commercial enterprise.

4. If yes, does a tourist boycott due to local poaching threat?
See above comment.

5. How effective are laws on wildlife crime, for example The Department of Justice's Environment and Natural Resources Division (ENRD), together with United States Attorneys are responsible for prosecuting international wildlife trafficking crimes, primarily under the Endangered Species Act (ESA) and the Lacey Act.
We believe our background regarding how reducing protections affect poaching speaks to the effectiveness of laws such as the ESA (in our context) and specific rules (such as the 10(j)). The finding that full protections seem to consistently reduce the incidence of poaching relative to reduced protection (less strict laws) suggests some effectiveness at mitigating anthropogenic mortality.

6. Is there any impact on international laws on wildlife conservation?
To our knowledge, international laws do not impact the conservation of the red wolf or other US wolf populations, as it should not given the intent of the ESA (i.e., to conserve species within the nation regardless of their international status).

7. Does harvests made without complying with the regulations for legal harvest result in the illegal taking of wildlife and come under poaching, where only wildlife can be poached.

Regarding the red wolf, there have never been any harvest seasons given its status as an experimental population of a critically endangered species. For hunters of coyotes, the killing of a red wolf would be illegal and we classify that as poaching (following Treves et al. 2017a,b, Agan et al. 2021). This classification is independent of intent, so we classify it as poaching regardless of whether the hunter turns themselves in or claims a mistaken identity of the victim. The latter is consistent with the Congressional intent and plain meaning of the ESA that judges killing an endangered species without a permit to be illegal regardless of intent or knowledge.

8. As it happens elsewhere, was there any impact of poverty that led to animals poaching?

We do not know of any research investigating the link between poverty and poaching of red wolves in North Carolina, USA. However, this seems unlikely given lack of evidence of any illegal trade in red wolf parts and very few economic value of red wolves, if any. Moreover, hunters in the area are predominately suburbanites who travel in to hunt (they are not local). Poverty in the area is largely restricted to the Black and Hispanic communities, who largely do not hunt.

9. It would be interesting to investigate number of animals are poached annually?

Agan et al (2021), Table 1 provides such numbers for reported poached as well as disappeared individuals ('FU') and notes the main cause of death in red wolves is poaching. The intent of our research is to analyze how changes in policy may affect the monitoring history of wolves.

10. The current wave of poaching is carried out by sophisticated and well-organized criminal networks – using helicopters, night-vision equipment, tranquilizers and silencers to kill animals at night, avoiding law enforcement patrols. Are the laws effective to curtail these aspects to protect carnivores and prey bases?

This may be the case internationally with species of high market value. However, that is not the case in the US and red wolves, as we are dealing with a species that reflects no market value, is limited in numbers and restricted to a relatively small area. See also comments on items 5, 7 and 8 for relevant clarifications.

11. Line 10: illegal killing (poaching) should be one of the causes for deaths of carnivores

We agree and note that we write is it the major cause of death for large carnivores in several regions (not overall or the only one).

12. Use 'Threatened species' rather endangered (IUCN standard term) line26

We did not make this change because the red wolf is also listed as critically endangered by the IUCN, and thus 'endangered species' seems more accurate. 'Threatened' species has a lower priority (than 'endangered') under the ESA and may confuse readers.

13. One or two lines should be included on result output in the Abstract.

We agree and will certainly include those at the RR Stage 2. For now we would rather not include outputs in the abstract, as this does not seem to be required for a Stage 1 RR.

14. Line 43: I suggest poaching is one of the many causes

Please see our response to item 11 above.

15. Line: 50 use threatened instead of endangered

We have added 'threatened' to the sentence.

16. Line 91-96: how legally killing associated with fewer disappearances?

The mentioned evidence from Suutarinen & Kojola seems to suggest increasing harvest quotas may increase poaching, while actual wolves killed decreases poaching. However, we cite Treves et al. in the text following that discussion to note concerns over their analysis and the potential for legal killing being associated with less poaching because wolves were killed legally before they could be poached (and that such relationship would demand competing risk approaches, such as ours).

17. Line 119-1122: It is little contradictory when Mexican gray wolves' deaths and disappearances un controlled condition and reduced federal protections and higher disappearances.

We have edited the explanation to make this clearer starting on line 119: The conditions under which the Mexican wolves were analyzed are described as more controlled given the the lack of change in lethal removals by agency personnel during different protection periods.

18. Line 147-158: if hybridization between coyotes and wolves defined as one the major threats, was the mitigation was effective? While the captures from the wild extirpated the species in 1996, and further gunshot deaths of red wolves in later years put pressure in later years. Does the Reintroduction was successful?

Hybridization was successfully managed, especially since it is caused by excessive human-caused mortality. We have included relevant language and citations. As for the Reintroduction Program, given our research is only focusing on the association between red wolf mortality and management policies, we do not think it appropriate to pass judgment on the reintroduction program in this manuscript, although our results will certainly inform how management affects mortality.

19. Line 164-180: when it is difficult to distinguish coyotes and red wolves and constantly killing either of these, how the reintroduction program effective? It would be interesting to look at this aspect.

It would certainly be so, and we believe we will address that question alongside Agan et al (2021) by looking at hazards and incidences of red wolves between different periods of coyote hunting that were more/less restrictive.

20. Line 200-212: Bases on various studies, many times, the collared individuals persist in suitable areas, while the radio collars stop working, wild population may vary and disappearances of species as well...

Indeed, disappearances may vary randomly and radio collars may fail. However, we would argue that, if disappearance risk is mostly due to these causes, the risk should be similar across the policy and management periods we consider (especially within the 6 alternating coyote hunting periods), since policies should not affect the risk of collar failure or non-poaching disappearances. Consider also the limited population size and geographic extent of the RWRA (5 counties), which mitigates confounding disappearances due to emigration and allows for more intensive monitoring (weekly flights and annual trapping) of the population, increasing detection and recollaring in cases of collar failure. We also account for population trend (increasing, stable, decreasing) through our management phase covariate.

21. Line 214-221: It is interesting to note that poaching is 28.3% and disappeared individuals was 22%, when considered point 20, it would result differently.

Which substantiates the FWS claim of monitoring collared wolves frequently and carefully because in most wolf populations studied thus far (e.g., Wisconsin wolves, Mexican wolves) LTF > poaching. Also, adding the two together means disappearances and poaching accounted for the majority of collared wolves, which seems to us to emphasize the need for FWS to investigate LTF and poaching cases, and their last locations, carefully. We thank the reviewer for reminding us of important points for discussion

22. Line 236-250: based on point 19, when coyotes were hunted day and night and people might be taking down red wolves, how agencies protect the species.

State/federal agencies did not respond with increased enforcement during those periods when coyote hunting was allowed (see Coalition v. N.C. Wildlife Res. Comm'n, No. 2:13-CV-60-BO (E.D.N.C. May. 13, 2014)). Rather, coyote hunting was prohibited by a court injunction the first two times. The last, ongoing period seems to hope to protect red wolves by allowing coyote hunting in certain areas and subject to certain requirements (see lines 259-261).

23. Table 4 the analysis and interpretation on hypotheses are rightly depicted.
Thank you.

24. I think it is a great studies and need to be published.

Thank you for your encouragement, clarifying questions and constructive feedback.

Appendix B

29 March 2022

Dear editors of *Royal Society Open Science*,

Please consider our submission entitled “**Evaluating how management policies affect red wolf mortality and disappearance**” (Manuscript ID RSOS-210400) as a Stage 2 registered report for *Royal Society Open Science*. We are grateful for the opportunity to complete the registered report process. This article received in-principle acceptance (IPA) at *Royal Society Open Science* on 8 July, 2021. Following IPA, the accepted Stage 1 version of the manuscript was preregistered on the OSF (URL) on 14 July 2020 (link on our Stage 2 submission). This preregistration was performed prior to data analysis. As per the requirements for a Stage 2 registered report submission, we have shared our raw and processed data to Dryad (links on our Stage 2 submission).

As proposed on our Stage 1 report, our finalized study evaluates the effects of various policy and management interventions on the hazard and incidence of illegal killing of a U.S. endangered species, the red wolf (*Canis rufus*), in the wild in northeastern North Carolina, USA. We use advanced biostatistical methods to analyze hazard, competing risks, and cumulative incidence functions for both disappearance (‘lost to follow-up’) and mortality endpoints, following recent studies, including a recent RSOS registered report [1,2]. We include disappearance as an endpoint alongside known-fate deaths and compare endpoint rates in different, replicated policy periods. The latter technique is novel in wildlife science and transformative because it takes advantage of new insights that have falsified the assumption that marked animals that disappear can be ‘censored’ without biasing results.

We hereby certify that for the primary registered report we did not analyze the data beyond what was present in the registered report Stage 1 and that the proposed analysis began immediately upon provisional acceptance of the Stage 1 manuscript in July 2020 and OSF protocol registration. We have made changes to the Stage 1 manuscript methods text and tables to correct for minor changes in our dataset as we prepared it for analysis and removed some observations. Further minor wording edits have been made on the introduction and methods to ensure clarity and that consistent language was used throughout the manuscript. These edits are all included as track changes. We certify that no further changes have been made.

Thank you for considering and with kind regards,

Francisco J. Santiago-Ávila, Suzanne Agan, Joseph W. Hinton, and Adrian Treves

References

1. Santiago-Ávila FJ, Chappell RJ, Treves A. 2020 Liberalizing the killing of endangered wolves was associated with more disappearances of collared individuals in Wisconsin, USA. *Sci. Rep.* , 1–14. (doi:10.1038/s41598-020-70837-x)
2. Louchouart N, Santiago-Ávila FJ, Parsons DR, Treves A. 2021 Evaluating how lethal management affects poaching of Mexican wolves. *R. Soc. Open Sci.* **8**. (doi:https://doi.org/10.1098/rsos.200330)

Appendix C**ROYAL SOCIETY
OPEN SCIENCE****Evaluating how management policies affect red wolf
mortality and disappearance**

Journal:	Royal Society Open Science
Manuscript ID	RSOS-210400.R2
Article Type:	Registered Report - Stage 2
Date Submitted by the Author:	29-Mar-2022
Complete List of Authors:	Santiago-Ávila, Francisco; University of Wisconsin-Madison; Project Coyote; The Rewilding Institute Agan, Suzanne; Kennesaw State University, Department of Ecology, Evolution, and Organismal Biology Hinton, Joseph W.; Wolf Conservation Center Treves, Adrian; University of Wisconsin-Madison
Subject:	ecology < BIOLOGY, environmental science < BIOLOGY
Keywords:	conservation, endangered species, poaching, survival analysis, large carnivore, Canis rufus
Subject Category:	Ecology, Conservation, and Global Change Biology

Author-supplied statements

Relevant information will appear here if provided.

Ethics

Does your article include research that required ethical approval or permits?:

This article does not present research with ethical considerations

Statement (if applicable):

CUST_IF_YES_ETHICS :No data available.

Data

It is a condition of publication that data, code and materials supporting your paper are made publicly available. Does your paper present new data?:

Yes

Statement (if applicable):

The pre-registered Stage 1 report can be found on the Open Science Framework at the following link: https://osf.io/wk6ay/?view_only=d3649ffc01834bcfabfbb2fa8017fd51.

The raw dataset from the U.S. Fish and Wildlife Service Red Wolf Recovery Program has been submitted to Dryad and can be found at the following link:

https://datadryad.org/stash/share/ujdCABtW6DgYt3P_WwYuV0aryLNrhX1zTa_KsYIGZAo. We have also included the prepared and processes data in our Dryad submission, which is ready to be run through the STATA code included starting on Supplemental Materials.

Conflict of interest

I/We declare we have no competing interests

Statement (if applicable):

[revised manuscript text omitted]

Wisconsin gray wolves [from 11] (see Table S4 for all parameters). BFs strength of evidence
was interpreted as follows: $1/3 < \text{BF} < 3$ would be inconclusive evidence; $\text{BF} > 3$ would
represent substantial evidence for the alternative hypothesis; $\text{BF} < 1/3$ would represent
substantial evidence for the null hypothesis of no association.

BF Specifications	Reported Poached		LTF	
	HR	SHR	HR	SHR
Mexican gray wolves	7.87	25.46	0.47	0.62
WI gray wolves	6.46	15.62	0.95	1.09

**Data Accessibility**

The pre-registered Stage 1 report can be found on the Open Science Framework at the following
link: https://osf.io/wk6ay/?view_only=d3649ffc01834bcfabfbb2fa8017fd51. The raw dataset
from the U.S. Fish and Wildlife Service Red Wolf Recovery Program has been submitted to
Dryad and can be found at the following link:

https://datadryad.org/stash/share/ujdCABtW6DgYt3P_WwYuV0aryLNrhX1zTa_KsYIGZAO

We have also included the prepared and processes data in our Dryad submission, which is ready
to be run through the STATA code included starting on Supplemental Materials.

**Competing Interests**

FSA, SA and JH declare no competing interests. AT declares no competing interests, and
provides his CV (http://faculty.nelson.wisc.edu/treves/archive_BAS/Treves_vita_Jan2020.pdf)
and all funding awarded as of 6 Jan 2020
(http://faculty.nelson.wisc.edu/treves/archive_BAS/funding.pdf) for transparency, so readers can
decide if they perceive a competing interest.

**Author Contributions**

All authors developed the study. The stage 1 manuscript was drafted by FSA and was edited and
reviewed by all authors. Data analyses were conducted by FSA, and independent interpretation
of the results was first conducted by SA, JH, and AT. Final interpretation and discussion of
results was conducted by all authors. The discussion was drafted by FSA, with edits and reviews
conducted by all authors. All authors have approved the final version of the manuscript.

1
2
3 746 **Acknowledgments**
4

[revised manuscript text omitted]
*) in the North Carolina Red Wolf Non-Essential Experimental population released in North
Carolina, USA from 1987-2020 (n=526). ~~Specifically, w~~We evaluated how changes in federal
and state policies protecting critically endangered red wolves in northeastern North Carolina,
USA, influenced the ~~Here, we report on a test of the opposed hypotheses that the hazard and~~
~~incidence of mortality and disappearance of critically endangered red wolves.~~ We observed
substantial increases in the hazard and incidence of red wolf reported poaching, and smaller
increases in disappearances, during periods of reduced federal and state protections (including
liberalizing coyote hunting of coyotes, *C latrans*, a species of similar appearance to wolves) for
wolves; white-tailed deer (*Odocoileus virginianus*) and American black bear (*Ursus*
*americanus*) hunting seasons; and management pPhases. ~~Similarly, we observed associated~~
~~increases of hazard and incidence of red wolf disappearance for all our policy, hunting season,~~
~~and management covariates, albeit of smaller magnitude.~~ Observed increases in hazard (85–
256%) and incidence of reported being poached (56–243%) support the ‘facilitated illegal

33 ~~‘killing hypothesis’ that predicts by which poachers killed more red wolves when federal and~~
~~state protections were relaxed for red wolves.~~ ~~wolves and coyotes (*Canis latrans*), a species of~~
~~similar appearance to wolves. (*Canis rufus*) in the Red Wolf Recovery Area, northeastern North~~
34 ~~Carolina, USA were affected by changes in federal and state policies that directly or indirectly~~
~~reduced protections for the species. We employ survival and competing risk models for data on~~

[revised manuscript text omitted]

reduced protections legally [11]. Next, Louchouart et al. [10] analyzed deaths and disappearances
of endangered Mexican gray wolves (*C. l. baileyi*) in Arizona and New Mexico, USA (1998-
2016) under even more statistically-controlled conditions: the hazard and incidence of
government-implemented removals of Mexican wolves did not change during the same periods,
eliminating any potentially confounding effects of super-additive mortality and undermining the
notion that the government reduced protections so they could act (i.e., remove wolves) to raise

tolerance [10]. They found periods of reduced federal protections for the subspecies were
associated with ~~relatively higher~~ increases of 121% in the hazard of disappearances. As in the
Wisconsin study [11] [cite], the disappearances also over-compensated ~~ing~~ for minimal declines in
incidence of reported poaching. Therefore, Louchouart et al. [10] ~~[cite]~~ dismissed the ‘killing
for tolerance’ hypothesis and added support in favor of both the ‘facilitated illegal killing’
(increased poaching) and ‘facilitated cryptic poaching’ (increased *cryptic* poaching) hypotheses.
Indeed, the latter study suggests that a government policy signal without attendant government
action to remove or kill wolves sufficed to elevate poaching and shift it from reported to cryptic.

[revised manuscript text omitted]

Resources Commission (NCWRC) liberalized coyote hunting in the 5-county RWRANC
recovery area [37,40,45,49,51], and (ii) the USFWS recovery program issued red wolf ‘take’
permits under the ESA 10(j) rule (Table 2). The NCWRC allowed daytime hunting of coyotes
throughout North Carolina on July 1, 1993 (Personal communication from Brian C. Daniel, Jan
21st, 2021). Following Aug 1, 2012, coyote hunting subsequently went through six alternating
policy periods, starting with n. ~~Nighttime hunting of coyotes was permitted on private lands by~~
~~the NCWRC on Aug 1, 2012, which but was~~ ended on Nov 20, 2012, because of a preliminary
injunction by a federal court to prevent red wolf mortality [48]. On July 26, 2013, the NCWRC
again authorized nighttime hunting of coyotes in the RWRANC recovery area which ended on
May 12, 2014, when a US District court again issued an injunction (*Coalition v. N.C. Wildlife*
*Res. Comm'n, No. 2:13-CV-60-BO)* ~~citation~~. Finally, ~~from on~~ Feb 27, 2015 through the present,
coyote hunting was conditionally permitted in the RWRANC recovery area during the daytime,
~~with permit and reporting requirements. This is the current policy for coyote hunting in the~~
RWRA.

To the periods of reduced state protections, we add a period of reduced federal ESA protections
~~for wolves~~ during which the agency issued permits to landowners for the take of red wolves.
Since 1995, red wolves have been managed under the 10(j) rule (section 10(j) of the ESA),
which allows for the issuance of such permits. However, the USFWS did not issue such permits
until 2014, and did not have to revise the 10(j) rule to do so. Thus, the agency issuance of
permits reflects a change in practice rather than ESA policy. The USFWS first issued a take
permit for red wolves on February 6, 2014. This agency action was then challenged in court on

November 2015, culminating in a preliminary injunction preventing the take of red wolves on
September 28, 2016, later made permanent on November 5, 2018 [51].
_____
We modeled ~~R~~recovery ~~P~~rogram (~~RWRP~~) phases as a categorical variable (*mgmt_phase*) to
control for management changes and population trends, following [40] (Table 3). Phase I ('1',
start of program – federal fiscal year 1998) represents the period when the ~~recovery~~
~~program~~Recovery Program focused its efforts on establishing a wild red wolf population [32].
Phase II ('2', fiscal years 1999-2005) represents the period when the ~~recovery program~~Recovery
~~Program~~ -began implementing the Red Wolf Adaptive Management Plan (~~RWAMP~~) to manage
hybridization with coyotes [35]. Phase III ('3', ~~October 1 fiscal year~~ 2006 – June 29, 2015)
represents the period when the ~~recovery program~~Recovery Program staff attempted to address
state coyote management as a response to both stagnant or declining wolf population and
increased rates of anthropogenic mortality [36,37,45]. Lastly, Phase IV ('4', June 30, 2015 – end
of analysis) represents the period during which the USFWS halted red wolf reintroductions while
continuing with removals, pending an evaluation of the recovery program [48]. ~~Federal fiscal~~
~~years for USFWS start 1 October and end 30 September the following year.~~
_____
Lastly, we modeled yearly hunting period(s) as a binary variable (*hunt_period*) identifying the
fall and winter hunting periods for ~~white-tailed deer, American~~ black bear, ~~deer~~ and waterfowl
('1', Sept 12 – Jan 31; '0' otherwise; Table 4) following [46], given evidence of increased
anthropogenic mortality during said periods.
_____

Other variables are unlikely to confound the effect of policy interventions because a variable
would have to co-occur in a majority of the periods of reduced protection and then change states
in multiple periods of stricter protection for red wolves in addition to being widespread across
the RWRANC recovery area. We searched USFWS recovery program documents, reports, and
the scholarly literature and found no evidence of other unaccounted for human or nonhuman
changes likely to strongly influence red wolf mortality cyclically or intermittently throughout the
RWRANC recovery area (e.g., periods of disease or significant changes in environmental
conditions).

*Statistical Methods*

We constructed hazard and subhazard models following a competing risk framework, which is
an extension of survival ('time-to-event') analyses [58], following the methods in [10,11].

Survival analyses focus on the estimation of 'time-to-event' functions, i.e., the probability of
observing a time interval (T), from the beginning of monitoring to an endpoint, greater than
some stated value t, $S(t) = P(T > t)$, within a specified analysis time (our study period, 1987-March
1, 2020)). Additionally, the same techniques allow for calculating the (endpoint-specific) hazard
function, $h_k(t)$; the instantaneous rate of occurrence of a particular endpoint k conditional on not
experiencing any endpoint until that time [59–62]. Cox models estimate covariate effects on the
endpoint-specific hazard as $h_k(t) = h_{0k}(t)e^{(\beta_1x_1 + \dots + \beta_jx_j)}$, where $h_{0k}(t)$ is an unestimated
baseline hazard function (i.e., semi-parametric) and β_j represents estimates of hazard ratios
(HRs) for each covariate x_j (HR<1 indicates a reduction in hazard, and HR>1 an increase in
hazard; Fig 1). We employed Lunn & McNeil's data augmentation method (by k endpoints)
along with a stratified (by endpoint) joint Cox proportional hazards model to simultaneously

estimate endpoint-specific changes in hazard ratios (HRs) for each endpoint-covariate interaction
 [63].

Figure 1. Cumulative hazard functions (CIFs) for 32 cases of legal killing taken from a sample of
 513 collared Wisconsin wolves [11]. Lines show cumulative hazard over monitoring time of
 'legal killing' endpoint derived from a univariate Cox model for policy periods of strict
 protections (i.e., baseline hazard for the 32 wolves killed legally, blue line) relative to policy
 periods of liberalized killing (red line; reflecting an HR=3.30 over the baseline hazard for the 32
 wolves).

However, a limitation of only relying on Cox models and hazard rates is that they do not
 consider competing risks. Competing risk analyses extend standard survival analyses by

366 simultaneously considering multiple endpoints (e.g.: multiple causes of death, agency removal,
and LTF). These analyses allow for the estimation of changes in incidence (i.e., the relative
incidence) of a particular endpoint conditional on a covariate level, while accounting for the
hazard of experiencing other competing endpoints. In this framework, an individual can
experience the end of monitoring time by only one of multiple mutually exclusive endpoints,
each associated with a particular probability ('risk'), so we refer to the endpoints as 'competing'
over time to bring about that event.

_____
Instead of hazards, competing risk techniques estimate the endpoint-specific cumulative
incidence function (CIF), defined by the failure probability $\text{Prob}(T < t, D = k)$; that is, the
*cumulative probability* of endpoint k (element D is an index variable that specifies *which*
*endpoint* occurred) occurring first at time T (*when* the event happened) within the study period
interval defined over time t in the presence of other competing endpoints [58,62,64].

_____
Within the competing risks framework, Fine-Gray (FG) subhazard models allow for estimating
differences ~~the estimation of differences~~ in CIFs for a given endpoint conditional on covariates
[64,65]. FG models use regression techniques similar to Cox models, except the interpretation of
model parameters changes: covariate estimates are interpreted as subhazard ratios (SHRs rather
than Cox's HRs), or relative incidence (instead of relative risk/hazard), in the presence of other
endpoints (i.e., for each covariate x_j : $\text{SHR} < 1$ indicates a reduction in incidence, and $\text{SHR} > 1$ an
increase in incidence). Despite both being informative and complementary, competing risk
models consider more information and provide greater predictive power [62,64,66].

_____

In this study, we exploit the complementarity of both models: our joint stratified Cox model
allowed us to test the hypothesis that our management covariates affected the *rate of occurrence*
(i.e., hazard) of specific endpoints, and endpoint-specific FG models allowed us to test if and
how much these same covariates affected the *probability* and *incidence* of said endpoints. The
endpoint-specific CIFs derived from these models allowed us to visualize any effects on
incidence while considering the prevalence of each endpoint in the population (Fig 2). Therefore,
we used both hazard and incidence to infer the changes due to our covariates and test our
hypotheses (Table 1, columns A-D). As a last step, we compare CIFs derived from both models
(Cox or FG) visually to assess consistency in model results and identify the most appropriate CIF
model [67].

Figure 2. Cumulative incidence functions (CIFs) for 513 collared Wisconsin wolves [11]. Lines
show separate endpoints for lost to follow up, LTF (n=243, orange), reported poached (n=88,

maroon), and legal kills (n=32, black) in two periods, derived from Fine Gray models for MAIN
imputation scenario. For each endpoint, we illustrate the cumulative incidence for liberalized
killing periods (dashed lines) and periods of full protection (solid lines).

For both our joint stratified Cox proportional hazards and FG subhazard models, we
assume both endpoint and time-to-endpoint for each wolf (subject) is independent of other
wolves' (i.e., one wolf's monitoring history and endpoint do not inform others') and that
censoring is independent of other endpoints (since we also account for LTF as an endpoint) [61],
given evidence that mortality of a paired red wolf breeder is usually followed by subsequent
pairing (not mortality) of the surviving breeder [40]. Since we split the monitoring history of
each wolf into multiple spells to include time-dependent covariates, we cluster all analyses by a
unique wolf identifier [68] to account for auto-correlation. We also evaluate compliance with the
proportionality assumptions of our Cox models using Schoenfeld residuals [59–61], and control
for such non-proportionality when necessary in both models through the inclusion of time-
varying coefficients (tvc). Both statistical models also comply with the appropriate number of
events per variable (EPV; 10>EPV for Cox, 40-50 EPV for FG) recommended in the scientific
literature to ensure accurate estimation of regression coefficients and their associated quantities
for the main endpoints of interest (~~LTF and poached~~ reported poached and LTF) (see Table 2)
[69–71]. We selected our best Cox and FG models based on parsimony, Akaike's Information
Criterion (AIC), Bayesian Information Criterion (BIC) and compliance with proportionality
assumptions for each model. In doing so, we avoided including covariates unless they are
essential to control for. We visually assessed goodness of fit for our Cox model using a Cox-

Snell residual plot [59,61]. We conducted all statistical analyses in Stata 16 (StataCorp LLC,
College Station, TX, 2019).

_____
We followed recommendations for rigorous competing risk analysis [62,64,66,72], reporting
results on all endpoint-specific hazards, incidences and CIFs, synthesizing them to elucidate
interactions between endpoints, but with a focus on anthropogenic causes (legal, reported
poached, LTF). For example, separate analyses of Wisconsin and Mexican wolves suggested
increases in both the hazard and incidence of LTF during periods of reduced ESA protections
overcompensated for smaller decreases in hazard and incidence of reported wolf-poaching
estimated during those same periods [10,11]-(Fig 2 above).

_____
We assessed the strength of evidence for any observed changes in hazard and incidence with
Bayes' Factors (BF) [73], as in [10]. We used Dienes' free online BF calculator, found at
http://www.lifesci.sussex.ac.uk/home/Zoltan_Dienes/inference/Bayes.htm. Dienes' BF calculator
assumes parameter estimates are normally distributed with known variance, which applies well
to the model coefficients obtained from Cox and FG models (for HRs and SHRs, respectively).
Following the discussion by Louchouart et al. [10] on BF specifications, we assume a half-
normal function with an expected standard deviation of SD=point estimate [73,74]. In specifying
these SDs, we follow [73]'s recommendations to select a most likely value while remaining blind
to the data, as requested of a registered report. We used endpoint-specific estimates from the
policy treatment variable from the Wisconsin and Mexican wolf populations [10,11] to model the
SD of the BF distributions.

_____

We report BFs for our main endpoints of interest; i.e., reported poached and LTF. We interpreted
the BF strength of evidence for each hypothesis (or null) following [73,74]: $1/3 < \text{BF} < 3$ –
inconclusive evidence; $\text{BF} > 3$ – substantial evidence for the alternative hypothesis; $\text{BF} < 1/3$ –
substantial evidence for the null hypothesis (i.e., no effect).

Table 1. Relationship between our hypotheses, proposed analyses and interpretation of outcomes (including contingent interpretation
 and synthesis of model results). Cox PH refers to Cox proportional hazards models, while Fine & Gray refers to competing-risk
 models. HR_{poa} refers to the hazard ratio of the reported poaching endpoint, while HR_{lrf} refers to the hazard ratio of the LTF endpoint
 (table modified from Table 4 in [10]). Define PH and remind reader of what Cox refers to

A. Question	B. Hypotheses	C. Analysis Plan	D. Interpretation given different outcomes †
Does the hazard or incidence of death by reported poaching or disappearance of wild, collared adult red wolves change after policies change from more to less protection and back again?	'Killing for tolerance' predicts the hazard and incidence decline for the endpoint 'poached' or the endpoint LTF of reported poaching (POA) and disappearances (LTF) decline when reducing protections for the species.	Cox PH models (for each endpoint) on policy and individual covariates. Competing risk Fine & Gray models (for each endpoint) on policy and individual covariates. CIFs allow for analysis of population effects (incidence) while considering the prevalence of each endpoint in the population.	(HR _{poa} has to be <1 and greater in magnitude than any increase in HR _{lrf} OR HR _{lrf} has to be <1 and greater in magnitude than any increase in HR _{poa}) AND endpoint-specific CIFs estimate which endpoint has a greater effect on the population. The criterion for determining if TOTAL poaching probability for wolves declined is a decline in the combined incidence of the LTF and POA endpoints.
	'Facilitated illegal killing' predicts the hazard and incidence of reported poaching (POA) and disappearances (LTF) increase increase for the endpoint 'poached' (POA) or the endpoint LTF when reducing protections for the species.	Cox PH models (for each endpoint) on policy and individual covariates. Competing risk Fine & Gray models (for each endpoint) on policy and individual covariates. CIFs allow for analysis of population effects (incidence) while considering the prevalence of each endpoint in the population.	(HR _{poa} has to be >1 and greater than any decrease in HR _{lrf} OR HR _{lrf} has to be >1 and greater than any decrease in HR _{poa}) AND endpoint-specific CIFs estimate which endpoint has a greater effect on the population (from Fine-Gray models of competing risks). The criterion for determining if TOTAL poaching probability for wolves declined increased is an increase decline in the combined incidence of the LTF and POA endpoints.

	'Facilitated cryptic poaching' predicts the hazard and incidence increase for the endpoint LTF when reducing protections for the species.	Cox PH models (for LTF endpoint) on policy and individual covariates. Competing risk Fine & Gray models (for LTF endpoint) on policy and individual covariates. LTF CIFs allow for analysis of population effects (incidence) while considering the prevalence of each endpoint in the population.	$HR_{ltf} > 1$ AND endpoint-specific CIFs estimate the effect on the population (from Fine-Gray models of competing risks).
--	---	--	---

455 † Bayes Factors (BFs) estimates the strength for our alternative and null hypotheses for each endpoint of interest, as well as assess
inconclusiveness of the data.

Table 2. Number of endpoints (unique wolf IDs) during periods of reduced state or federal
 protections ('1') in the RWRANC recovery area (1986-1987-March 1, 2020). Periods of reduced
 protections include state policies liberalizing coyote hunting in the RWRANC recovery area as
 well as federal issuance of take permits under the 10(j) rule. Wolves that survived to the end of
 the study period are omitted here and censored at the end of the study period (n=3).

Endpoint	Periods of state and federal policy for red wolves (red_prot)		Total
	0 strict protection	1 reduced protection	
Agency removal	4338	32	4640
Collision	5756	1212	6968
LTF	101	16	117
Nonhuman	6260	66	6866
PoachedReported poached	110109	4141	151150
Unknown	65	17	82
Total	438429	9594	533523
Time at risk (t=days)	490,212467,686	74,42473,410	564,636541,096

Table 3. Number of events (unique wolf IDs) per endpoint and ~~recovery program~~RWRP
 management phase (I-IV), following Hinton et al. [40] (~~1986~~1987-March 1, 2020, see *Methods*).
 Wolves that survived to the end of the study period are omitted here and censored at the end of
 the study period (n=3).

Endpoint	Red wolf recovery program phases (mgmt_phase)				Total
	Phase I	Phase II	Phase III	Phase IV	
	(1)	(2)	(3)	(4)	
Agency removal	19 18	22 18	54 54	00 00	4640 4640
Collision	19	17 16	25	8	6968 6968
LTF	21	37	53	6	117
Nonhuman	20	25 24	21 20	22 22	6866 6866
Poached Reported poached	11	39 38	86	15	151150 151150
Unknown	18	15	38	11	82
Total	108 107	155 148	228 226	4242 4242	533523 533523
Time at risk (t=days)	123,941 108, 440	161,209 157, 141	251,098 247, 607	28,388 27,90 8	564,636 541, 096

Table 4. Number of events (unique wolf IDs) per endpoint by fall-winter hunting season ('1', '0'
 otherwise; ~~1986~~1987-March 1, 2020). Wolves that survived to the end of the study period are
 omitted here and censored at the end of the study period (n=3).

Endpoint	Fall-winter hunting season (hunt_period)		Total
	0 No hunting	1 Hunting	
Agency removal	25 22	21 18	46 40
Collision	48	21 20	69 68
LTF	55	62	117
Nonhuman	37 36	31 30	68 66
Poached Reported poached	45	106 105	151 150
Unknown	59	23	82
Total	269 265	264 258	533 523
Time at risk (t=days)	350,332 334,920	214,304 206,176	564,636 541,096

**Timeline**

~~Our anticipated timeline for completion following in-principle acceptance and OSF registration~~

~~was 5-6 weeks to conduct all statistical analyses and 4-5 weeks for writing and submission.~~

**Results**

[revised manuscript text omitted]

Wisconsin gray wolves [from 11] (see Table S4 for all parameters). BFs strength of evidence
was interpreted as follows: $1/3 < \text{BF} < 3$ would be inconclusive evidence; $\text{BF} > 3$ would
represent substantial evidence for the alternative hypothesis; $\text{BF} < 1/3$ would represent
substantial evidence for the null hypothesis of no association.

BF Specifications	Reported Poached		LTF	
	HR	SHR	HR	SHR
Mexican gray wolves	7.87	25.46	0.47	0.62
WI gray wolves	6.46	15.62	0.95	1.09

**Data Accessibility**

The pre-registered Stage 1 report can be found on the Open Science Framework at the following
link: https://osf.io/wk6ay/?view_only=d3649ffc01834bcfabfbb2fa8017fd51. The raw dataset
from the U.S. Fish and Wildlife Service Red Wolf Recovery Program has been submitted to
Dryad and can be found at the following link:

https://datadryad.org/stash/share/ujdCABtW6DgYt3P_WwYuV0aryLNrhX1zTa_KsYIGZAo

We have also included the prepared and processes data in our Dryad submission, which is ready
to be run through the STATA code included starting on Supplemental Materials.

**Competing Interests**

FSA, SA and JH declare no competing interests. AT declares no competing interests, and
provides his CV (http://faculty.nelson.wisc.edu/treves/archive_BAS/Treves_vita_Jan2020.pdf)
and all funding awarded as of 6 Jan 2020
(http://faculty.nelson.wisc.edu/treves/archive_BAS/funding.pdf) for transparency, so readers can
decide if they perceive a competing interest.

**Author Contributions**

All authors developed the study. The stage 1 manuscript was drafted by FSA and was edited and
reviewed by all authors. Data analyses were conducted by FSA, and independent interpretation
of the results was first conducted by SA, JH, and AT. Final interpretation and discussion of
results was conducted by all authors. The discussion was drafted by FSA, with edits and reviews
conducted by all authors. All authors have approved the final version of the manuscript.

1
2
3 **820 Acknowledgments**
4

[revised manuscript text omitted]

1041